# Fine-Tuning Discrete Diffusion Models with Policy Gradient Methods

**Oussama Zekri**
CREST, ENSAE
Institut Polytechnique de Paris
France
oussama.zekri@ensae.fr

**Nicolas Boullé**
Department of Mathematics
Imperial College London
United Kingdom
n.boulle@imperial.ac.uk

## Abstract

Discrete diffusion models have recently gained significant attention due to their ability to process complex discrete structures for language modeling. However, fine-tuning these models with policy gradient methods, as is commonly done in Reinforcement Learning from Human Feedback (RLHF), remains a challenging task. We propose an efficient, broadly applicable, and theoretically justified policy gradient algorithm, called Score Entropy Policy Optimization (SEPO), for fine-tuning discrete diffusion models over non-differentiable rewards. Our numerical experiments across several discrete generative tasks demonstrate the scalability and efficiency of our method. Our code is available at https://github.com/ozekri/SEPO.

## 1 Introduction

Diffusion models have become efficient generative modeling tools in various tasks, including image and video generation (Song et al., 2021; Ho et al., 2020). Although most of the applications of diffusion models depend on a continuous state space (such as images), recent works extended these models to discrete settings, enabling their use in language modeling and other discrete generative tasks (Sun et al., 2023; Campbell et al., 2022; Austin et al., 2021; Benton et al., 2024). Moreover, several studies showed that these models can be competitive with autoregressive models, such as GPT (Brown et al., 2020) or Llama (Touvron et al., 2023), while allowing for more flexible generation compared to next-token prediction (Lou et al., 2024; Sahoo et al., 2024; Shi et al., 2024). These discrete diffusion models are promising if they can be scaled up to natural language processing tasks.

However, fine-tuning discrete diffusion models is challenging. Existing strategies, such as classifier guidance (Ho and Salimans, 2021; Nisonoff et al., 2025; Gruver et al., 2024) and steering (Rector-Brooks et al., 2025), often face scalability issues or intractable objectives. We focus instead on reinforcement learning (RL), which optimizes a pre-trained model via rewards. A key challenge is that sampling from categorical distributions is non-differentiable and incompatible with standard gradient methods. Recent work (Wang et al., 2025) addresses this using the Gumbel-Softmax trick (Jang et al., 2017), but requires differentiable rewards and becomes memory-intensive at scale. In contrast, policy gradient methods like PPO (Schulman et al., 2017) and GRPO (Shao et al., 2024) rely only on reward evaluations and offer stability, unbiased gradients, and robustness to noise.

We propose a policy gradient algorithm, Score Entropy Policy Optimization (SEPO), designed for discrete diffusion. Unlike Wang et al. (2025), our method handles non-differentiable rewards, enabling broader fine-tuning scenarios. SEPO offers a unified and scalable framework for optimizing discrete diffusion models, that supports both conditional and unconditional generation. Other policy gradient approaches have also been proposed, such as GLIDE (Cao et al., 2025), which relies on reward shaping, and d1 (Zhao et al., 2025), specifically designed for masked diffusion models and cannot

39th Conference on Neural Information Processing Systems (NeurIPS 2025).

handle unconditional generation. These methods are tailored to specific setups, whereas `SEPO` offers a more general and principled framework for discrete diffusion fine-tuning.

**Main contributions.**

1) We provide an explicit characterization of policy gradient algorithms for discrete diffusion models in the concrete score framework. This allows the use of non-differentiable rewards in discrete fine-tuning tasks *without* steering or guidance mechanisms, for both **conditional** and **unconditional** generation.

2) We propose an efficient, scalable algorithm based on policy gradient methods (Schulman et al., 2017; Shao et al., 2024), called Score Entropy Policy Optimization (`SEPO`), for discrete diffusion. It uses a clipped-ratio loss with **self-normalized importance sampling** for low-variance. We also introduce a gradient flow alternative that improves sample quality at a higher complexity.

3) We perform numerical experiments on DNA fine-tuning and natural language tasks to demonstrate the performance of our methods. We achieve state-of-the-art DNA results: **Pred-Activity** 7.64 and **ATAC-Acc** 99.9%, topping prior RL and guidance baselines with lower run-to-run variance.

## 2 Background and preliminaries

### 2.1 Related works

**Inference-time techniques.**   Inference-time techniques are simple yet effective as they require no fine-tuning or training when reward functions are available. Recent studies (Singhal et al., 2025; Ma et al., 2025) showed that they can achieve competitive performance by scaling computational resources. Although inference-time techniques offer distinct advantages, they typically result in longer inference times compared to fine-tuned models. The key considerations for these techniques include computational efficiency and differentiability of the reward (Uehara et al., 2025).

**Policy gradients algorithms.**   Policy gradient algorithms are a key class of reinforcement learning methods that optimize parameterized policies by directly maximizing expected returns. Modern implementations include Proximal Policy Optimization (Schulman et al., 2017) or Group Relative Policy Optimization (Shao et al., 2024). These algorithms are highly sensitive to policy design since the architecture impacts expressiveness, optimization stability, and exploration.

**Fine-tuning diffusion models with Reinforcement Learning.**   In the case of continuous diffusion models, fine-tuning via policy gradients has been proposed (Fan et al., 2024; Li et al., 2024; Black et al., 2024; Ren et al., 2025). In a more recent study, (Marion et al., 2024) implements `REINFORCE` algorithm (Williams, 1992) for continuous diffusion models in a single-loop algorithm, avoiding nested optimization. However, extending these approaches to discrete diffusion models is more challenging. This work adapts these studies to the discrete case and extends them to general policy gradient algorithms.

### 2.2 Discrete Diffusion

In discrete diffusion models, the dynamics of a single particle is described by a continuous-time Markov chain (CTMC), denoted as a stochastic process $(x_t)_{0 \leq t \leq T}$ operating on a finite space $\mathcal{X} = \{\alpha_1, \ldots, \alpha_m\}^n$. Here, $(\alpha_i)_{1 \leq i \leq m}$ represents the possible states that form a vocabulary of size $m$, and $n$ is the length of the sequences, which is a fixed number known as *context window* or *block size*. Typically, it describes sequences of tokens or image pixel values. While the size $d := |\mathcal{X}| = m^n$ of $\mathcal{X}$ is exponential in $n$, deep neural networks such as transformers (Vaswani, 2017) were shown to perform and generalize well on these incredibly large state spaces (Zekri et al., 2024).

**Forward process.**   At any given time $t$, the distribution of a particle $x_t$ is given by $\boldsymbol{p}_t$, which lies within the probability simplex $\Delta_d \subset \mathbb{R}^d$. The forward process is a noising process that maps the initial data distribution $\boldsymbol{p}_0 := p_{\text{data}}$ to some final noisy distribution $\boldsymbol{p}_T := p_{\text{ref}}$, which is easy to sample. During the noising forward process, the particle's probability transitions between states are given by a rate matrix $Q_t \in \mathbb{R}^{d \times d}$, indexed by $\mathcal{X}$, through the equation $\frac{\mathrm{d}\boldsymbol{p}_t}{\mathrm{d}t} = Q_t \boldsymbol{p}_t$ for $t \in [0, T]$.

The time reversal of this equation (Kelly, 2011) is known as,

$$\frac{d\boldsymbol{p}_{T-t}}{dt} = \overline{Q}_{T-t}\boldsymbol{p}_{T-t}, \quad t \in [0, T], \tag{1}$$

where for $x, y \in \mathcal{X}, \overline{Q}_t(y, x) = \begin{cases} \frac{\boldsymbol{p}_t(y)}{\boldsymbol{p}_t(x)} Q_t(x, y), & x \neq y, \\ -\sum_{z \neq x} \overline{Q}_t(z, x), & x = y. \end{cases}$

**Score entropy.** Lou et al. 2024 recently showed that one can approximate Eq. (1) via score entropy, inspired by concrete score matching (Meng et al., 2022). This is done by learning the concrete score as $s_\theta(x, t)_y \approx \boldsymbol{p}_t(y)/\boldsymbol{p}_t(x)$ with a sequence-to-sequence neural network $s_\theta$ parametrized by $\theta \in \mathbb{R}^p$. The resulting process is described by the following equation:

$$\frac{d\boldsymbol{q}_t^\theta}{dt} = \overline{Q}_{T-t}^\theta \boldsymbol{q}_t^\theta, \quad t \in [0, T], \tag{2}$$

where the denoising process $\boldsymbol{q}_t^\theta \approx \boldsymbol{p}_{T-t}$ maps $\boldsymbol{q}_0^\theta := p_{\text{ref}}$ to $\boldsymbol{q}_T^\theta$, and $\theta$ is learned to achieve $\boldsymbol{q}_T^\theta \approx_\theta p_{\text{data}}$. The matrix $\overline{Q}_t^\theta$ is defined for $x, y \in \mathcal{X}$ as $\overline{Q}_t^\theta(x, y) = s_\theta(x, t)_y Q_t(x, y)$ if $x \neq y$ and $\overline{Q}_t^\theta(y, y) = -\sum_{z \neq x} \overline{Q}_t^\theta(z, y)$. In practice, the quantity $s_\theta(x, t)_y$ is available for all $y \in \mathcal{X}$ at Hamming distance (Hamming, 1950) one of $x$: the states $y$ that differ from $x$ by exactly one token. This represents only $\mathcal{O}(mn)$ ratios instead of $\mathcal{O}(m^{2n})$ (Campbell et al., 2022; Lou et al., 2024).

**Masked diffusion models.** The score entropy setup also contains the simplified frameworks detailed in (Sahoo et al., 2024; Shi et al., 2024; Ou et al., 2024), often referred to as masked diffusion models. These models define a simplified training loss based on a weighted cross-entropy objective by introducing a special [MASK] token, and have been shown to outperform uniformly noised discrete diffusion models. Promising attempts at scaling them to larger settings support this observation (Arriola et al., 2025; Nie et al., 2025; Ye et al., 2025).

## 2.3 Fine-tuning with Reinforcement Learning

After the pretraining phase, a discrete diffusion model with learned parameter $\theta_{\text{pre}}$ aims to approximate $p_{\text{data}}$, in the sense $\boldsymbol{q}_T^{\theta_{\text{pre}}} \approx p_{\text{data}}$. Our goal is to fine-tune the denoising process $\boldsymbol{q}_t^\theta$ to increase a reward function $R_t : \mathcal{X} \rightarrow \mathbb{R}$, without having access to $p_{\text{data}}$.

**Minimization problem.** We focus on optimization problems over implicitly parameterized distributions. For a given family of functions $(\mathcal{F}_t)_{t \in [0, T]} : \Delta_d \rightarrow \mathbb{R}$, we aim to minimize the loss function

$$\ell_t(\theta) := -\mathcal{F}_t(\boldsymbol{q}_t^\theta), \quad t \in [0, T], \tag{3}$$

over $\theta \in \mathbb{R}^p$. Standard choices of $\mathcal{F}_t$ include $\mathcal{F}_t(\boldsymbol{q}_t^\theta) = \mathbb{E}_{x \sim \boldsymbol{q}_t^\theta}[R_t(x)]$ to maximize a reward function $R_t : \mathcal{X} \rightarrow \mathbb{R}$, or $\mathcal{F}_t(\boldsymbol{q}_t^\theta) = -\text{KL}(\boldsymbol{q}_t^\theta \| \boldsymbol{q}_t^{\theta_{\text{pre}}})$ to minimize the KL divergence of $p_t$ from a distribution $\boldsymbol{q}_t^{\theta_{\text{pre}}}$. As detailed in (Uehara et al., 2024), a typical fine-tuning algorithm for diffusion models combines these two terms as

$$\ell_t(\theta) = -\mathbb{E}_{x \sim \boldsymbol{q}_t^\theta}[R_t(x)] + \alpha \, \text{KL}(\boldsymbol{q}_t^\theta \| \boldsymbol{q}_t^{\theta_{\text{pre}}}), \tag{4}$$

where $\alpha > 0$ is a weighting factor. One can recast the denoising process as a finite-horizon multi-step Markov Decision Process formulation with known transition probabilities as follows:

$$\text{For all } t \in \{0, \dots, T\}, \quad \mathcal{S} = \mathcal{X}, \quad \mathcal{A} = \mathcal{X}, \quad \text{Policy } \boldsymbol{q}_t^\theta,$$
$$S_t = x_{T-t}, \quad A_t = x_{T-1-t}, \quad \mathbb{P}(S_0) = p_{\text{ref}}, \quad \mathbb{P}(S_{t+1} \mid S_t, A_t) = \delta(\{S_{t+1} = A_t\}),$$

where $\mathcal{S}$ and $\mathcal{A}$ denote the states and action spaces, $S_t$ and $A_t$ denote the states and actions, and $\delta$ is the Dirac delta distribution. Although common choices in the fine-tuning diffusion models with reinforcement learning literature (Black et al., 2024; Fan et al., 2024; Clark et al., 2024; Uehara et al., 2024) often set $R_t = 0$ for $t < T$ and $R_T = R$, we retain the general form of $R_t$ in the results for greater generality. This choice allows for more flexible reward evaluation schemes. For example, one can run fewer denoising steps and stop the process at some intermediate time $T_0 < T$, assigning the reward $R_{T_0} = R$ based on the partially denoised sample. We will consider this case, and the corresponding distribution $\boldsymbol{q}_{T_0}^\theta$ will be denoted by $\boldsymbol{\pi}_\theta$.

**Loss reward gradient.** To apply first-order optimization methods, one needs to compute the gradient $\nabla_\theta \ell_t(\theta)$. Since $\mathcal{X}$ is a finite space of size $d$, we have

$$\nabla_\theta \ell_t(\theta) = -\nabla_\theta \big( \mathcal{F}_t(\boldsymbol{q}_t^\theta) \big) = -\mathbf{f}^{\mathrm{T}} \nabla_\theta \boldsymbol{q}_t^\theta, \tag{5}$$

where $\mathbf{f} \in \mathbb{R}^d$ is the vector of first variations $\mathcal{F}_t(\boldsymbol{q}_t^\theta)$ (see Appendix D.1). Importantly, we note that $\mathbf{f}(\boldsymbol{p})(x) = R_t(x)$ for $x \in \mathcal{X}$, which does not involve the differentiability of $R_t$ (with respect to some embedding $\mathcal{E}(\mathcal{X})$ of the state space). One can then design deterministic non-differentiable functions that act on $\mathcal{X}$ as rewards, similar to those arising in RLHF, or elsewhere. This may include designing desired protein properties (Rector-Brooks et al., 2025).

## 3 Methods

### 3.1 Policy gradients for concrete score

The gradient of the target distribution $\nabla_\theta \boldsymbol{q}_t^\theta$ in Eq. (5) can be approximated based on its relationship with the concrete score $s_\theta$.

**Assumption 3.1.** *For $x \neq y \in \mathcal{X}$, define the exact ratio $r_{x,y}^t(\theta) := \boldsymbol{q}_t^\theta(y)/\boldsymbol{q}_t^\theta(x)$. We assume that there exists $\delta > 0$ such that for all $\theta \in \mathbb{R}^p$, $x \neq y \in \mathcal{X}$, $t \in [0, T]$,*

$$\| \nabla_\theta \log s_\theta(x, T - t)_y - \nabla_\theta \log r_{x,y}^t(\theta) \| \leq \delta. \tag{6}$$

The following theorem shows that one can compute $\widehat{\nabla \ell_t}(\theta)$ that estimates $\nabla_\theta \ell_t(\theta)$ through a discrete analogue of the `REINFORCE` algorithm (Williams, 1992).

> **Theorem 3.1** (Discrete `REINFORCE` trick). *Under Assumption 3.1 and with the notations introduced in Section 2, applying the `REINFORCE` with concrete score approximations yields the following estimator of Eq. (5):*
>
> $$\widehat{\nabla \ell_t}(\theta) = \sum_{x \in \mathcal{X}} \boldsymbol{q}_t^\theta(x) R_t(x) \sum_{y \neq x} \boldsymbol{q}_t^\theta(y) \nabla_\theta \log s_\theta(x, T - t)_y,$$
>
> *such that $\| \widehat{\nabla \ell_t}(\theta) - \nabla_\theta \ell_t(\theta) \| \leq R_{\max} \delta$, where $R_{\max} := \sup_{t \in [0,T]} \max_{x \in \mathcal{X}} |R_t(x)| < \infty$.*

**Computation of the loss function.** The summand in Theorem 3.1 involves the unknown distributions $\boldsymbol{q}_t^\theta(x)$ and $\boldsymbol{q}_t^\theta(y)$. While the outer sum can be estimated via Monte Carlo sampling (with $N$ samples $(x_1, \ldots, x_N)$), the inner sum is weighted by $\boldsymbol{q}_t^\theta(y)$. As noted in Lou et al. (2024), a single $x \in \mathcal{X}$ provides access to every component of the concrete score $s_\theta(x, t)_y$, for $y \neq x$, and then to $\boldsymbol{q}_t^\theta(y \mid x)$ since this is the only missing quantity in the sampling scheme (Eq. (11) in Appendix A). Hence, one may be tempted to estimate $\boldsymbol{q}_t^\theta(y) = \sum_{x \in \mathcal{X}} \boldsymbol{q}_t^\theta(y \mid x) \boldsymbol{q}_t^\theta(x) \approx \frac{1}{N} \sum_{i=1}^N \boldsymbol{q}_t^\theta(y \mid x_i)$. However, as illustrated in Fig. 1, for a given $y$, we only have access to a single value $\boldsymbol{q}_t^\theta(y \mid x_i)$; we do not even have two such values, let alone $N$, and thus cannot take advantage of the full sample set used in the initial Monte Carlo estimation, leading to a really high-variance estimation. In fact, for any reasonable block size $n$, although it is certain that at least one sample $x_i$ will be a neighbor of $y$, it is highly unlikely that two distinct samples $x_i$ and $x_j$ both have $y$ as a neighbor, since this would require them to differ from $y$ at exactly one token position.

**Self-normalized importance sampling (SNIS) of $\boldsymbol{q}_t^\theta(y)$.** To address this issue, for a given $y$, we build $M$ samples $\{z_i\}_{1 \leq i \leq M}$ from $\boldsymbol{q}_t^\theta(\cdot \mid y)$ and employ the importance sampling approximation

$$\boldsymbol{q}_t^\theta(y) = \sum_{z \in \mathcal{X}} \boldsymbol{q}_t^\theta(y \mid z) \boldsymbol{q}_t^\theta(z) = \sum_{z \in \mathcal{X}} \boldsymbol{q}_t^\theta(y \mid z) \frac{\boldsymbol{q}_t^\theta(z)}{\boldsymbol{q}_t^\theta(z \mid y)} \boldsymbol{q}_t^\theta(z \mid y) \approx \frac{1}{M} \sum_{i=1}^M \boldsymbol{q}_t^\theta(y \mid z_i) \frac{\boldsymbol{q}_t^\theta(z_i)}{\boldsymbol{q}_t^\theta(z_i \mid y)},$$

with weights $w(z_i) = \boldsymbol{q}_t^\theta(z_i)/\boldsymbol{q}_t^\theta(z_i \mid y)$ and proposal $\boldsymbol{q}_t^\theta(z_i \mid y)$. Following Bayes' rule $\boldsymbol{q}_t^\theta(z_i \mid y) \propto \boldsymbol{q}_t^\theta(y \mid z_i) \boldsymbol{q}_t^\theta(z_i)$, the proposal distribution $\boldsymbol{q}_t^\theta(z_i \mid y)$ is **optimal** in the sense that it minimizes the variance of the importance weights. This makes it a theoretically sound and efficient choice for

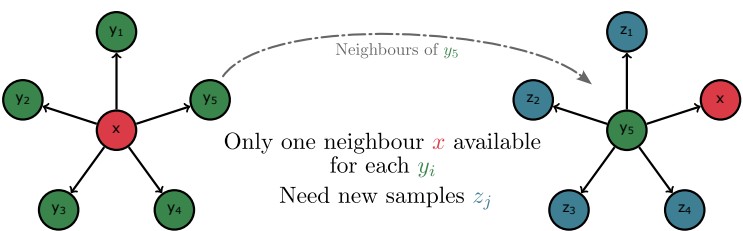

Figure 1: **On the estimation of $q_t^\theta(y)$. Left:** Given a sample $x$, only its neighbors $\{y_i\}_{1\le i\le 5} \in \mathcal{X}$ are accessible for computing $q_t^\theta(y_i \mid x)$, and each $y_i$ typically has only one such parent $x$ in the sampled batch. **Right:** For a given $y_5$, there are several neighbours $\{z_j\}_{1\le j\le 5}$. It is unlikely to find multiple distinct samples such that both are neighbors of $y_5$, since this would require them to differ from $y_5$ at exactly the same token position.

importance sampling. However, the marginal distribution $q_t^\theta(z_i)$ is typically intractable, so evaluating $w(z_i)$ is unfeasible. Fortunately, one has access to an approximation of $q_t^\theta(z_i)$ up to a normalization constant thanks to the score $1/s_\theta(z_i, T-t)_y \approx q_t^\theta(z_i)/q_t^\theta(y)$. This is sufficient for self-normalized importance sampling with weights $\tilde{w}(z_i) \propto q_t^\theta(z_i)/q_t^\theta(z_i \mid y)$. The estimator obtained via SNIS takes the form

$$\hat{q}_t^\theta(y) = \frac{\sum_{i=1}^M q_t^\theta(y \mid z_i)\tilde{w}(z_i)}{\sum_{i=1}^M \tilde{w}(z_i)} = \left(\frac{1}{M}\sum_{i=1}^M \frac{1}{q_t^\theta(y \mid z_i)}\right)^{-1}.$$

Thus, the SNIS estimate can be computed as the harmonic mean of the conditionals $q_t^\theta(y \mid z_i)$. The estimator is both consistent and asymptotically unbiased and has significantly lower variance than the naive single sample estimate. Under a mild moment assumption, we prove in Appendix C.2 that $\mathbb{V}(\hat{q}_t^\theta(y)) = \mathcal{O}(M^{-1})$. Finally, the gradient can be computed as $\nabla_\theta \ell_t(\theta) = \mathbb{E}_{x\sim q_t^\theta}[R_t(x)g(x,\theta)]$, where $g(x,\theta) := \sum_{\substack{y\in\mathcal{X} \\ y\ne x}} q_t^\theta(y)\nabla_\theta \log s_\theta(x, T-t)_y$, and we estimate $q_t^\theta(y)$ with the SNIS method described above.

**Importance sampling of the outer sum.** Although this defines an unbiased gradient, the REINFORCE algorithm is known to have high variance and to not restrict large policy updates. To address the latter limitation and estimate $g(x,\theta)$, we build upon the core ideas introduced by Trust Region Policy Optimization (TRPO) (Schulman et al., 2015a). Instead of sampling from $q_t^\theta$ one can leverage importance sampling through an old policy $q_t^{\theta_{old}}$, and constraint the KL divergence between the old and the current policy as follows:

$$\nabla_\theta \ell_t(\theta) = \mathbb{E}_{x\sim q_t^{\theta_{old}}}\left[R_t(x)\frac{q_t^\theta(x)}{q_t^{\theta_{old}}(x)}g(x,\theta)\right]. \tag{7}$$

**Assumption 3.2.** *We assume that there exists $\delta > 0, \rho > 0$ and $G > 0$ such that for all $\theta \in \mathbb{R}^p$, $x \ne y \in \mathcal{X}$, $t \in [0,T]$, $\|\log s_\theta(x, T-t)_y - \log r_{x,y}^t(\theta)\| \le \delta$ and $\|\nabla_\theta \log s_\theta(x, T-t)_y - \nabla_\theta \log r_{x,y}^t(\theta)\| \le \delta$, $q_t^\theta(x)/q_t^{\theta_{old}}(x) \le \rho$ and $\|\nabla_\theta \log s_\theta(x, T-t)_y\| \le G$.*

Once adapted for concrete score, this formulation leads us to the following result.

**Theorem 3.2** (Importance sampling gradient). *Under Assumption 3.2 and with the notations introduced in Section 2, applying the TRPO with concrete score approximations yields the following estimator of Eq. (5):*

$$\widehat{\nabla \ell_t}(\theta) = \mathbb{E}_{x\sim q_t^{\theta_{old}}}\left[R_t(x)h(x,\theta)\right], \tag{8}$$

*such that* $\|\widehat{\nabla \ell_t}(\theta) - \nabla \ell_t(\theta)\| \le R_{\max} C(\rho, G)\delta$, *where* $R_{\max} := \sup_{t \in [0,T]} \max_{x \in \mathcal{X}} |R_t(x)| < \infty$, $C(\rho, G) > 0$ *is a constant depending on* $\rho$ *and* $G$, *and*

$$h(x, \theta) = \sum_{\substack{y \in \mathcal{X} \\ y \ne x}} \boldsymbol{q}_t^\theta(y) \frac{\boldsymbol{q}_t^\theta(y)}{\boldsymbol{q}_t^{\theta_{\mathrm{old}}}(y)} \frac{s_{\theta_{\mathrm{old}}}(x, T - t)_y}{s_\theta(x, T - t)_y} \nabla_\theta \log s_\theta(x, T - t)_y.$$

The quantity $h(x, \theta)$ is expressed in this way in Eq. (7) to emphasize how the loss will be computed in practice. While being the founding step of state-of-the-art policy gradient algorithms, TRPO requires solving a constrained optimization problem at each step. However, thanks to Theorem 3.2, we can now build powerful, stable, scalable, and easy-to-implement policy gradient algorithms.

## 3.2 SEPO : Score Entropy Policy Optimization

Our algorithm relies on the ideas introduced in (Schulman et al., 2017; Shao et al., 2024), but can be adapted to any policy gradient algorithm built on REINFORCE or TRPO. Inspired from these algorithms, we clip $\boldsymbol{q}_t^\theta(x)/\boldsymbol{q}_t^{\theta_{\mathrm{old}}}(x)$ via the following ratio that appears in the inner sum of Theorem 3.2:

$$u_{x,y}^t = \frac{\boldsymbol{q}_t^\theta(x)}{\boldsymbol{q}_t^{\theta_{\mathrm{old}}}(x)} = \frac{\boldsymbol{q}_t^\theta(y)}{\boldsymbol{q}_t^{\theta_{\mathrm{old}}}(y)} \frac{s_{\theta_{\mathrm{old}}}(x, T - t)_y}{s_\theta(x, T - t)_y}.$$

to the interval $[1 - \epsilon, 1 + \epsilon]$ for some hyperparameter $\epsilon > 0$. This can be interpreted as enforcing a trust region condition analogous to $\rho = 1 + \epsilon$ in Assumption 3.2. Another advantage of discrete diffusion models is their great generation flexibility. It is then be possible to apply our algorithm **conditionally** (via a training dataset, typically in RHLF) or **unconditionally** for fine-tuning. Hence, in the conditional form, Eq. (7) becomes

$$\mathbb{E}_{z \sim \mathcal{D}} \mathbb{E}_{x \sim \boldsymbol{q}_t^{\theta_{\mathrm{old}}}(x|z)} \left[ R_t(x) \frac{\boldsymbol{q}_t^\theta(x \mid z)}{\boldsymbol{q}_t^{\theta_{\mathrm{old}}}(x \mid z)} g(x \oplus z, \theta) \right].$$

where $x \oplus z$ represents the concatenation of $x$ and $z$. Instead of using directly the reward $R_t(x)$, we compute an advantage $A(x)$ to reduce the variance of the Monte-Carlo estimations. This quantifies how much better an action is compared to the expected return at a given state. A common approach in PPO (Schulman et al., 2017) is to learn a value network to approximate the reward, and then define the advantage $A_t$ with Generalized Advantage Estimation (Schulman et al., 2015b). For GRPO (Shao et al., 2024), the advantage is the standardized reward over each group. Specifically, for a group of outputs $x = \{x_1, \ldots, x_G\}$, we have that $A_t(x_i) = \frac{R_t(x_i) - \mathrm{mean}(R_t(x))}{\mathrm{std}(R_t(x))}$, for $i \in \{1, \ldots, G\}$.

**Remark 3.3.** *The loss function takes the form*

$$\ell^A(\theta) = \mathbb{E}_{x \sim \boldsymbol{\pi}_{\theta_{\mathrm{old}}}} \Big[ \sum_{\substack{y \in \mathcal{X} \\ y \ne x}} w_{x,y} \log s_\theta(x, T - T_0)_y \Big], \tag{9}$$

*where* $w_{x,y} = \boldsymbol{\pi}_\theta(y) f(u_{x,y}^{T - T_0})$ *is a coefficient (where $f$ is a function to specify) and the log concrete score* $\log s_\theta(x, T - T_0)_y$ *is the only term with an attached gradient. PPO, GRPO, and other methods can be constructed by specifying the function $f$ in the coefficient $w_{x,y}$. In Appendix B, we present a unified framework encompassing methods that can be derived from* SEPO.

Optionally, for $t \in [0, T_0]$, a $\mathrm{KL}(\boldsymbol{q}_t^\theta \| \boldsymbol{q}_t^{\theta_{\mathrm{pre}}})$ term can also be added to the loss, as in Eq. (4). Although this is not absolutely necessary, as clipping already implicitly regularizes with a $\mathrm{KL}(\boldsymbol{q}_t^\theta \| \boldsymbol{q}_t^{\theta_{\mathrm{old}}})$ term (Schulman et al., 2017; Fan et al., 2024), the derivation is given in Appendix D.2, for completeness. This leads to the Score Entropy Policy Optimization (SEPO) algorithm described in Algorithm 1. SEPO iteratively samples from the target distribution via a CTMC (Line 4) and optimizes $\theta_s$ using an optimization objective (Line 6), refining the policy with policy gradients. Specifically, **Line 4** generates samples from the target distribution $\boldsymbol{\pi}_{\theta_{\mathrm{old}}}$ using the CTMC $\overline{Q}^{\theta_{\mathrm{old}}}$. This can be done in $O(1)$ time complexity by leveraging the queuing trick introduced in (Marion et al., 2024, Alg. 3), at a higher memory cost. **Line 6** updates the parameters $\theta_s$ using a policy optimization algorithm based on the objective $\ell^A$ (see Eq. (9)).

This means performing $K$ iterations of gradient ascent (or descent) on the policy loss function to improve the policy $\pi(\theta_s)$ using the previously collected samples and computed advantages.

### 3.3 Sampling through gradient flow

**Bilevel problem.** We use sampling to reach the limiting process of the backward distribution $q_t^\theta$. This procedure can be interpreted as optimizing a functional $\mathcal{G} : \Delta_d \times \mathbb{R}^p \to \mathbb{R}$ over $\Delta_d \subset \mathbb{R}^d$ as

$$\pi_\theta = \underset{\boldsymbol{p} \in \Delta_d}{\arg\min}\, \mathcal{G}(\boldsymbol{p}, \theta).$$

---

**Algorithm 1** SEPO

1: **Require**: CTMC $\overline{Q}^\theta$, iteration $S$, epoch $K$
2: Set $\theta_0$ and $\theta_{\text{old}}$ to $\theta_{\text{pre}}$
3: **for** $s \in [1, \cdots, S]$ **do**
4:     Sample from $\pi_{\theta_{\text{old}}}$ with $\overline{Q}^{\theta_{\text{old}}}$
5:     Compute the reward and the advantage
6:     Optimize $\theta_s$ with $\ell^A$ for $K$ epochs
7:     Set $\theta_{\text{old}}$ to $\theta_s$
8: **end for**
9: **Output**: $\theta_{S+1}$

---

When $\pi_\theta$ is the limiting distribution of an infinite-time process (e.g., Langevin diffusion in the continuous case, Langevin 1908; Pavliotis 2014), one can recast Eq. (3) as a bilevel optimization problem. This has been proposed by Marion et al. 2024 in the continuous case and allows to efficiently alternate between optimizing one-step of the inner problem and one-step of the outer problem.

**Gradient flow interpretation.** In our case, $\pi_\theta$ is reached with *finite-time horizon*, in $T$ steps of sampling. However, it is possible to reach $\pi_\theta$ in *infinite-time horizon* by sampling from a specific time-homogeneous CTMC. The choice of the functional $\mathcal{G}(\boldsymbol{p}, \theta) = \mathrm{KL}(\boldsymbol{p} || \pi_\theta)$ leads to a gradient flow interpretation of sampling via a specific CTMC.

> **Lemma 3.3** (Gradient flow). *Sampling from the following ordinary differential equation*
>
> $$\frac{\mathrm{d}\boldsymbol{p}_t}{\mathrm{d}t} = Q_t^c \boldsymbol{p}_t, \quad \text{where } Q_t^c := Q_t + \overline{Q}_t,$$
>
> *implements a gradient flow for* $\mathrm{KL}(\,\cdot\,|| \boldsymbol{p}_t)$ *in* $\Delta_d$, *with respect to a Wasserstein-like metric.*

**Corrector steps.** Of course, $s_\theta$ is not perfectly learned in practice, and we just have access to the rate matrix $Q_t^{c,\theta} := Q_t + \overline{Q}_t^\theta$. But this gives us insight into the choice of our sampling strategy, especially with predictor-corrector techniques for discrete diffusion, introduced in (Campbell et al., 2022) and developed in (Zhao et al., 2024). We will then sample from the time-homogeneous CTMC of rate $Q_{T-T_0}^{c,\theta}$ to reach $\pi_\theta$ with infinite-time horizon. Note that this does not require computing an integral compared to the time-inhomogeneous case. We are then optimizing a functional in Wasserstein space through sampling (Marion et al., 2024; Bonet et al., 2024). Sampling from $Q_t^c$ affects Line 4 of Algorithm 1. In practice, the sample quality can be improved by adding corrector steps with $Q_t^c = Q_t + \overline{Q}_t^\theta$, as proposed in (Campbell et al., 2022). Once the process has run for $T_0$ steps, multiple sampling iterations from $Q_{T-T_0}^c$ can be performed.

**Linear system characterization.** In this case, $\nabla_\theta \pi_\theta$ in Eq. (5) will be obtained by solving a linear system, using the implicit function theorem (see Appendix C.4) on $\nabla_1 \mathcal{G}$, through a corrected version denoted $\nabla_\theta^\eta \pi_\theta$. While both the evaluation of the derivatives and the inversion of this linear system can be done automatically (Blondel et al., 2022), it is costly given the dimensionality $d$. Instead, we provide the exact linear system as well as a closed form of the inverse in Proposition 3.4. Note that this affects Line 6 of Algorithm 1, where $\nabla_\theta \pi_\theta$ in Eq. (5) is replaced by $\nabla_\theta^\eta \pi_\theta$.

> **Proposition 3.4.** *For each* $\eta > 0$, $\nabla_\theta^\eta \pi_\theta$ *is the solution to a linear system of the form*
>
> $$A_\eta \boldsymbol{X} = B_\eta \in \mathbb{R}^{d \times p},$$
>
> *where* $A_\eta$ *is a rank-1 update to the* $d \times d$ *identity matrix, whose inverse can be explicitly computed using the Sherman–Morrison formula.*

## 3.4 Convergence bounds

From a high-level point of view, Algorithm 1 alternates between sampling and optimization steps. We can then view Algorithm 1 as the following coupled equations:

$$\begin{aligned} \mathrm{d}\boldsymbol{q}_s &= Q_{T-T_0}^{c,\theta_s}\boldsymbol{q}_s\mathrm{d}s, \\ \mathrm{d}\theta_s &= -\beta_s\Gamma(\boldsymbol{q}_s,\theta_s)\mathrm{d}s, \end{aligned} \tag{10}$$

for $0 \leq s \leq S$. The gradient used on line 6 of Algorithm 1 depends both on $\boldsymbol{q}_s$ and $\theta_s$, and we refer to it as $\Gamma$ (so that $\nabla_\theta \ell^A(\theta_s) = \Gamma(\boldsymbol{\pi}_{\theta_s}, \theta_s)$). To simplify the analysis, the evolution of both $\boldsymbol{q}_s$ and $\theta_s$ is done in continuous time flow, for some $s \in [0, S]$, with $S > 0$. Let $\|\cdot\|$ denote the Euclidean norm on $\mathbb{R}^p$. We reintroduce assumptions on $\boldsymbol{\pi}_\theta$ and $\Gamma$ made in (Marion et al., 2024).

**Assumption 3.4.** *There exists $C \geq 0$ such that for all $x \in \mathcal{X}$ and $\theta \in \mathbb{R}^p$, $\|\nabla_\theta \boldsymbol{\pi}_\theta(x)\| \leq C$. There exists $\varepsilon > 0$ such that for all $x \in \mathcal{X}$ and $\theta \in \mathbb{R}^p$, $\boldsymbol{\pi}_\theta(x) > \varepsilon$.*

This assumption states that the gradient of the target distribution is bounded. The second part is similar to the ambiguity of the language often considered when studying models acting on spaces like $\mathcal{X}$ (Zekri et al., 2024; Hu et al., 2024; Xie et al., 2021).

**Assumption 3.5.** $\exists C_\Gamma \geq 0$ *such that for all $p, q \in \Delta_d$, $\theta \in \mathbb{R}^p$, $\|\Gamma(p,\theta) - \Gamma(q,\theta)\| \leq C_\Gamma\sqrt{\mathrm{KL}(p\|q)}$.*

This assumption essentially states that the gradient $\Gamma$ is Lipschitz with respect to the KL divergence on $\Delta_d$. With these in place, we establish the convergence of the average objective gradients.

> **Theorem 3.5** (Convergence of Algorithm 1). *Let $S > 0$ and $\theta_s$ be the solution to Eq. (10) with $\beta_s = \min(1, 1/\sqrt{s})$, for $s \in [0, S]$. Under Assumptions 3.4 and 3.5, we have, as $S \to \infty$,*
>
> $$\frac{1}{S}\int_0^S \|\nabla\ell^A(\theta_s)\|^2\mathrm{d}s = \mathcal{O}\left(1/\sqrt{S}\right).$$

# 4 Experiments

## 4.1 Language modeling

We briefly present our experiments on language modeling; see Appendix E.1 for details.

**Training.** We fine-tune the SEDD Medium Absorb model (Lou et al., 2024) using `SEPO` in an Actor-Critic PPO framework, directly optimizing with reinforcement learning without a supervised finetuning stage. The reward model, built from GPT-2 architecture and trained on the HH-RLHF dataset (Bai et al., 2022), provides reward signals to guide fine-tuning (see Figs. 4 and 5). We train two variants, SEDD-SEPO-128 and SEDD-SEPO-1024, respectively trained with $T = 128$ and $T = 1024$ denoising steps, to measure the impact of those denoising steps on final response quality.

**Quantitative evaluation.** We evaluate our models using pairwise comparisons with GPT-3.5 Turbo (Brown et al., 2020) as a judge over 153 prompts from the Awesome Chat-GPT Prompts dataset (Akın, 2023). As shown in Table 1, both SEDD-SEPO variants outperform the pretrained model, demonstrating the benefits of reinforcement learning fine-tuning. Increasing the number of denoising steps further improves response quality, with SEDD-SEPO-1024 achieving the best results overall. Qualitative results are deferred to Appendix E.1.

| | SEDD-SEPO-128 | | | SEDD-SEPO-1024 | | |
|---|---|---|---|---|---|---|
| Number of steps $T$ | 128 | 512 | 1024 | 128 | 512 | 1024 |
| SEDD Vanilla | 71.2% | 64.1% | 67.9% | 74.5% | **75.8**% | 73.2% |
| SEDD-SEPO-128 | × | × | × | 63.1% | **68.8**% | 67.8% |
| SEDD-SEPO-1024 | 36.9% | 31.2% | 32.2% | × | × | × |

Table 1: Proportion of outputs deemed favorable by the Judge LLM for each model and each denoising steps $T \in \{128, 512, 1024\}$. **Best** results are highlighted.

## 4.2 DNA sequence modeling

We present an experiment on a DNA sequence modeling task, which follows the setup of Wang et al. (2025). Additional details are provided in Appendix E.2.

**Dataset, settings, and baselines.** We employ the pretrained model of Wang et al. (2025), a masked discrete diffusion model (Sahoo et al., 2024) trained on $700k$ regulatory DNA sequences (200 bp) from the Gosai dataset (Gosai et al., 2023). Each sequence is annotated with a continuous enhancer activity score in the HepG2 cell line, and a reward model trained on this score is used during fine-tuning. Our DNA experiments are conducted in a fully *unconditional* setting: sampling starts from a fully noisy sequence and proceeds via denoising without any prefix or conditioning signal; rewards are computed only on the final denoised sample (i.e., with $T_0 = T$). We compare our method to a diverse set of state-of-the-art baselines: *(i) Guidance-based models*, including classifier guidance (CG) for discrete diffusion models (Nisonoff et al., 2025), Sequential Monte Carlo methods (SMC, TDS) (Wu et al., 2023), and classifier-free guidance (CFG) (Ho and Salimans, 2021); *(ii) Direct reward optimization* with DRAKES (Wang et al., 2025), which uses reward gradients in an RL-based fine-tuning setup for discrete diffusion models; *(iii) Policy Optimization* with GLID$^2$E (Cao et al., 2025), a gradient-free RL-based tuning method.

Table 2: Evaluation of DNA modeling methods. State-of-the-art performance is in **bold**, and the second-highest performance is underlined. Means over 3 seeds; standard deviations in parentheses.

| Method | Pred-Activity (median) ↑ | ATAC-Acc (%) ↑ | 3-mer Corr↑ | Log-Lik (median) ↑ |
|---|---|---|---|---|
| Pretrained | $0.17\,(0.04)$ | $1.5\,(0.2)$ | $-0.061\,(0.034)$ | $-261.0\,(0.6)$ |
| CG | $3.30\,(0.00)$ | $0.0\,(0.0)$ | $-0.065\,(0.001)$ | $-266.0\,(0.6)$ |
| SMC | $4.15\,(0.33)$ | $39.9\,(8.7)$ | $0.840\,(0.045)$ | $-259.2\,(5.5)$ |
| TDS | $4.64\,(0.21)$ | $45.3\,(16.4)$ | $0.848\,(0.008)$ | $-257.1\,(5.0)$ |
| CFG | $5.04\,(0.06)$ | $92.1\,(0.9)$ | $0.876\,(0.004)$ | $-265.0\,(2.6)$ |
| DRAKES | $5.61\,(0.07)$ | $92.5\,(0.6)$ | $\mathbf{0.887\,(0.002)}$ | $-264.0\,(0.6)$ |
| DRAKES w/o KL | $6.44\,(0.04)$ | $82.5\,(2.8)$ | $0.307\,(0.001)$ | $-281.6\,(0.6)$ |
| GLID$^2$E | $7.35\,(0.07)$ | $90.6\,(0.3)$ | $0.49\,(0.074)$ | $-239.9\,(1.4)$ |
| GLID$^2$E w/o M1 | $2.57\,(0.60)$ | $0.63\,(0.3)$ | $0.473\,(0.078)$ | $\mathbf{-239.1\,(10.1)}$ |
| GLID$^2$E w/o M2 | $6.62\,(0.42)$ | $67.3\,(39.4)$ | $0.458\,(0.009)$ | $-244.7\,(21.5)$ |
| SEPO | $7.55\,(0.01)$ | $99.5\,(0.2)$ | $0.500\,(0.004)$ | $-243.8(0.5)$ |
| SEPO with GF | $\mathbf{7.64\,(0.01)}$ | $\mathbf{99.9\,(0.09)}$ | $0.638\,(0.001)$ | $-248.2\,(0.1)$ |

**Evaluation metrics and results.** We report four metrics: (i) Pred-Activity, the median predicted activity from a held-out reward oracle; (ii) ATAC-Acc, a binary chromatin accessibility score in HepG2; (iii) 3-mer Corr, the correlation between 3-mer frequencies of generated and sequences from the Gosai dataset; and (iv) Log-Lik, the log-likelihood under the pretrained diffusion model. We find that SEPO **achieves the highest performance** across all key metrics except 3-mer Corr, confirming its ability to generate highly active and biologically relevant sequences while maintaining a reasonable alignment with the training distribution. Notably, SEPO with gradient flow (GF) achieves near-perfect chromatin accessibility ($99.9\%$) and the highest enhancer activity ($7.64$), surpassing both DRAKES and GLID$^2$E. While SEPO does not fully maximize 3-mer Corr, this metric primarily reflects low-level distributional similarity rather than functional relevance. Our results suggest that prioritizing enhancer activity and biological accessibility leads to more meaningful sequence generation in practical scenarios. Moreover, we observe that SEPO also exhibits significantly lower variance across runs (as shown by the small standard deviation in Table 2), highlighting the stability of our generated sequences, once the optimization process is performed.

Finally, as illustrated in Fig. 2, even in the worst case over $640$ samples, SEPO methods maintain a minimum Pred-Activity score around $6$, whereas all other methods exhibit much lower worst-case scores between $0$ and $2$. These strong performances, combined with the stability of the generation, make SEPO a highly effective counterpart to both inference-time methods and other RL-based fine-tuning approaches, providing a more reliable solution when generation quality is prioritized over sampling speed.

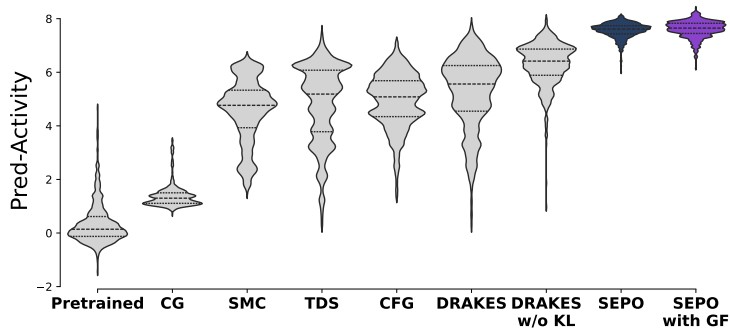

Figure 2: **Violin plot of Pred-Activity scores across models.** The plot shows the distribution of predicted enhancer activity (Pred-Activity) from the held-out reward oracle for each model, across $640$ generated sequences. SEPO and SEPO with GF achieve the highest Pred-Activity scores with low variance, illustrating the effectiveness as well as the stability of our optimization process. Results for $GLID^2E$ are not displayed as the corresponding finetuned model weights are not publicly available.

**Wall-clock and overhead.** SNIS adds a batched forward pass to compute transition probabilities for each of the $N$ outer samples, which moderately increases per-epoch cost but reduces the number of training epochs. On a single NVIDIA® GeForce RTX™ 3090 (24GB) GPU, we report the following computational timings.

| Method | KL Trunc. | #Epochs | Batch / GRPO | Runtime (hh:mm) | Time/Epoch (mm:ss) | Final Pred-Activity |
|---|---|---|---|---|---|---|
| SEPO | 10 | 14 | $8/8$ | 00:19 | 01:21 | 7.55 |
| SEPO with GF | 10 | 8 | $8/8$ | 00:24 | 02:59 | 7.64 |
| SEPO (no SNIS) | 10 | 199 | $8/8$ | 00:10 | 00:03 | 4.66 |
| DRAKES (no KL) | – | 127 | $8/-$ | 00:26 | 00:12 | 6.41 |
| DRAKES (with KL) | 50 | 66 | $8/-$ | 00:31 | 00:28 | 5.70 |

SEPO achieves higher final performance with fewer epochs despite a higher per-epoch cost, yielding competitive total runtime. Our SNIS estimator trades a forward pass for variance reduction, paying off in stability and wall-clock.

# 5   Conclusion

We introduced SEPO, a novel approach for fine-tuning discrete diffusion models using policy gradient methods. By extending previous work that applied these methods on continuous spaces, we developed a unified framework that adapts this methodology to the discrete case. SEPO is applicable to both **conditional** and **unconditional** generation tasks, broadening its applicability across diverse fine-tuning scenarios. Experimental results demonstrate the effectiveness of our approach in optimizing discrete diffusion models while addressing key challenges such as non-differentiability and combinatorial complexity. Future work includes further refining gradient estimation techniques and exploring applications in structured generative modeling. One limitation of the current method is that the inner importance sampling steps introduce additional computational complexity, as resampling is performed for each neighbor of the samples obtained from the outer Monte Carlo sampling.

# Acknowledgements

The authors would like to thank Pierre Marion, Anna Korba, and Omar Chehab for fruitful discussions. This work was supported by the Office of Naval Research (ONR), under grant N00014-23-1-2729. This work was done thanks to the ARPE program of ENS Paris-Saclay, which supported the visit of the first author to Imperial College London.

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

# Appendix

**Roadmap.**   In Appendix A, we first recall our notations and introduce additional definitions. Appendix B presents our unified paradigm of policy gradient methods, including PPO and GRPO. The detailed proofs of our theoretical results are given in Appendix C. Appendix D provides mathematical supplements and Appendix E reports additional experiments and implementation details.

# Table of Contents

# A  Additional background details

**Notations.**  This appendix uses the following notations: $[d]$ denotes the set of integers $\{1, \ldots, d\}$, $[t]_+ := \max\{0, t\}$, and $I_d$ stands for the $d \times d$ identity matrix. We also denote the diagonal matrix with diagonal components $(z_1, \ldots, z_d)$ by $\mathrm{diag}(z_1, \ldots, z_d)$. Finally, if $z = (z_1, \ldots, z_d)$, the same matrix will be denoted by $\mathrm{diag}(z)$.

**Sampling strategies.**  Sampling discrete diffusion models involves selecting efficient strategies to simulate the backward equation (2), while balancing computational cost and sample quality. Among other strategies for CTMCs, sampling can be done via the *tau-leaping* algorithm (Gillespie, 2001), which implements an Euler step at each position $i$ simultaneously and independently:

$$q_t(x_{t-\Delta t}^i | x_t^i) = \delta_{x_t^i}(x_{t-\Delta t}^i) + \Delta_t \overline{Q}_{T-t}^\theta(x_t^i, x_{t-\Delta t}^i) \tag{11}$$

Discrete diffusion models can also be used to perform flexible *conditional sampling* (Lou et al., 2024). Unlike *unconditional sampling*, which samples $q_t(x_{t-\Delta t} | x_t)$, we incorporate auxiliary data $\mathbf{c}$ by modifying the probability to be sampled to $q_t(x_{t-\Delta t} | x_t, \mathbf{c})$. Finally, the number of reverse diffusion steps, $T$, directly impacts computational efficiency and sample fidelity, with larger $T$ providing more accurate approximations of the target distribution at a higher computational cost.

# B  Unified Paradigm of Policy Gradient Methods

As in (Shao et al., 2024), we provide the expression of the coefficient $w_{x,y}$ for SEPO in its PPO and GRPO variants. This can be easily extended to the other training methods synthesized in Shao et al. 2024, Section 5.2.

## B.1  Proximal Policy Optimization (PPO)

For PPO (Schulman et al., 2017), the coefficient $w_{x,y}$ takes the form

$$w_{x,y}(\epsilon) = \pi_\theta(y) \min\{\mathrm{clip}(u_{x,y}^{T-T_0}, 1 - \epsilon, 1 + \epsilon)A(x); u_{x,y}^{T-T_0}A(x)\}, \quad \epsilon > 0.$$

A common approach is to learn a value network to approximate the reward, and then compute the advantage as $A(x) = R(x) - V(x)$.

## B.2  Group Relative Policy Optimization (GRPO)

For GRPO (Shao et al., 2024), we consider a group of outputs $x = \{x_1, \ldots, x_G\}$. The coefficient $w_{x,y}$ takes the form

$$w_{x,y}(\epsilon) = \frac{\pi_\theta(y)}{G} \sum_{i=1}^{G} \min\{\mathrm{clip}(r_{x_i,y}^{T-T_0}, 1 - \epsilon, 1 + \epsilon)A(x_i); r_{x_i,y}^{T-T_0}A(x_i)\}, \quad \epsilon > 0.$$

The advantages are the standardized reward over each group. Specifically, the advantages are defined as

$$A(x_i) = \frac{R(x_i) - \mathrm{mean}(R(x))}{\mathrm{std}(R(x))}, \quad i \in \{1, \ldots, G\}.$$

The KL term of the original GRPO objective (Shao et al., 2024) is discussed in Section 3.2.

# C  Proofs of the results

This section details the proof of the results that appear in the main text.

## C.1 Proof of Theorem 3.1

**Expression of $\nabla_\theta q_t^\theta$.** We begin by calculating $\nabla_\theta q_t^\theta$ appearing in Eq. (5) component-wise. Let $x \in \mathcal{X}$ and recall[1] that we can express $q_t^\theta(x)$ as $q_t^\theta(x) = \mathrm{e}^{-V_x(\theta)}/Z_\theta$ for some normalization constant $Z_\theta = \sum_{y \in \mathcal{X}} \mathrm{e}^{-V_y(\theta)}$. Therefore,

$$\nabla_\theta q_t^\theta(x) = -q_t^\theta(x)\nabla_\theta V_x(\theta) - \frac{\mathrm{e}^{-V_x(\theta)}\nabla_\theta Z_\theta}{Z_\theta^2} = -q_t^\theta(x)\nabla_\theta V_x(\theta) - q_t^\theta(x)\frac{\nabla_\theta Z_\theta}{Z_\theta}$$

$$= -q_t^\theta(x)\nabla_\theta V_x(\theta) - q_t^\theta(x)\left(\sum_{y \in \mathcal{X}}\frac{-\nabla_\theta V_y(\theta)\mathrm{e}^{-V_y(\theta)}}{Z_\theta}\right)$$

$$= -q_t^\theta(x)\nabla_\theta V_x(\theta) + q_t^\theta(x)\left(\sum_{y \in \mathcal{X}} q_t^\theta(y)\nabla_\theta V_y(\theta)\right).$$

**Plug-in estimator.** Define

$$\widehat{\nabla_\theta q_t^\theta}(x) := -q_t^\theta(x)\sum_{\substack{y \in \mathcal{X}\\ y \neq x}} q_t^\theta(y)\,\nabla_\theta \log s_\theta(x, T-t)_y.$$

Let the pointwise error be

$$\Delta(x) := \widehat{\nabla_\theta q_t^\theta}(x) - \nabla_\theta q_t^\theta(x) = -q_t^\theta(x)\sum_{\substack{y \in \mathcal{X}\\ y \neq x}} q_t^\theta(y)\Big(\nabla_\theta \log s_\theta(x, T-t)_y - \nabla_\theta \log r_{x,y}^t(\theta)\Big).$$

By (6) and since $\sum_{\substack{y \in \mathcal{X}\\ y \neq x}} q_t^\theta(y) \leq 1$,

$$\|\Delta(x)\| \leq q_t^\theta(x)\sum_{\substack{y \in \mathcal{X}\\ y \neq x}} q_t^\theta(y)\,\delta \leq q_t^\theta(x)\,\delta. \tag{12}$$

**Loss-gradient estimator and bias bound.** Recall $\ell_t(\theta) = -\sum_{x \in \mathcal{X}} R(x)q_t^\theta(x)$ and define

$$\widehat{\nabla \ell_t}(\theta) := -\sum_{x \in \mathcal{X}} R(x)\,\widehat{\nabla_\theta q_t^\theta}(x).$$

Then

$$\widehat{\nabla \ell_t}(\theta) - \nabla \ell_t(\theta) = -\sum_{x \in \mathcal{X}} R(x)\,\Delta(x).$$

Using (12) and that $\sum_x q_t^\theta(x) = 1$,

$$\left\|\widehat{\nabla \ell_t}(\theta) - \nabla \ell_t(\theta)\right\| \leq \sum_{x \in \mathcal{X}} |R(x)|\,\|\Delta(x)\| \leq \delta \sum_{x \in \mathcal{X}} |R(x)|\,q_t^\theta(x) \leq R_{\max}\,\delta.$$

**Conclusion.** Under the single approximation condition (6), the plug-in estimator obtained by replacing $\nabla_\theta \log r_{x,y}^t$ with $\nabla_\theta \log s_\theta$ in (5) yields the gradient formula stated in Theorem 3.1, and its bias with respect to the true $\nabla \ell_t(\theta)$ is bounded by $R_{\max}\delta$.

## C.2 Self-normalized importance sampling

We now formalize the self-normalized importance sampling (SNIS) estimator, which plays a key role in approximating marginal probabilities when direct computation is intractable. The following lemma establishes an explicit expression for the SNIS estimator of $q_t^\theta(y)$.

---

[1]Since we are working on a discrete finite state space $\mathcal{X}$, the normalization constant $Z_\theta = \sum_{y \in \mathcal{X}} \exp(-V_y(\theta))$ is finite as long as each $V_y(\theta) > -\infty$, which is the case.

**Lemma C.1.** *Let $y \in \mathcal{X}$ and $\{z_i\}_{1 \leq i \leq M}$ i.i.d. samples from $\boldsymbol{q}_t^\theta(\cdot \mid y)$. Define $\tilde{w}(z_i) \propto \frac{\boldsymbol{q}_t^\theta(z_i)}{\boldsymbol{q}_t^\theta(z_i|y)}$. Then, the SNIS estimator of $\boldsymbol{q}_t^\theta(y)$ is*

$$\hat{\boldsymbol{q}}_t^\theta(y) = \frac{\sum_{i=1}^M \boldsymbol{q}_t^\theta(y \mid z_i)\tilde{w}(z_i)}{\sum_{i=1}^M \tilde{w}(z_i)} = \left(\frac{1}{M}\sum_{i=1}^M \frac{1}{\boldsymbol{q}_t^\theta(y \mid z_i)}\right)^{-1}$$

*Proof.* By Bayes' rule, $\boldsymbol{q}_t^\theta(z_i \mid y) = \boldsymbol{q}_t^\theta(y \mid z_i)\boldsymbol{q}_t^\theta(z_i)/\boldsymbol{q}_t^\theta(y)$, so $\tilde{w}(z_i) = Z\frac{\boldsymbol{q}_t^\theta(z_i)}{\boldsymbol{q}_t^\theta(z_i|y)} = Z\frac{\boldsymbol{q}_t^\theta(y)}{\boldsymbol{q}_t^\theta(y|z_i)}$ where $Z$ is some constant independent of $z_i$. Therefore, $\sum_{i=1}^M \boldsymbol{q}_t^\theta(y \mid z_i)\tilde{w}(z_i) = MZ\boldsymbol{q}_t^\theta(y)$ and $\sum_{i=1}^M \tilde{w}(z_i) = Z\boldsymbol{q}_t^\theta(y)\sum_{i=1}^M \frac{1}{\boldsymbol{q}_t^\theta(y|z_i)}$. This directly yields that $\hat{\boldsymbol{q}}_t^\theta(y) = \left(\frac{1}{M}\sum_{i=1}^M \frac{1}{\boldsymbol{q}_t^\theta(y|z_i)}\right)^{-1}$. $\square$

Having obtained a closed-form expression for the SNIS estimator, we now investigate its statistical properties. In particular, the following result shows that its variance decreases at the canonical Monte Carlo rate.

**Lemma C.2.** *Let $y \in \mathcal{X}$ and $\{z_i\}_{1 \leq i \leq M}$ i.i.d. samples from $\boldsymbol{q}_t^\theta(\cdot \mid y)$. Define $U_i = \frac{1}{\boldsymbol{q}_t^\theta(y|z_i)}$ and assume that $\mathbb{E}[U_i^2] < \infty$. Then,*

$$\mathbb{V}(\hat{\boldsymbol{q}}_t^\theta(y)) = \mathcal{O}\left(\frac{1}{M}\right)$$

*Proof.* Define $U = \frac{1}{M}\sum_{i=1}^M U_i$ and $g(u) = 1/u$. One has $\hat{\boldsymbol{q}}_t^\theta(y) = g(U)$. We remark that $\mathbb{E}[U_i] = 1/\boldsymbol{q}_t^\theta(y)$ for each $1 \leq i \leq M$. Since $\mathbb{E}[U_i^2] < \infty$ by assumption, the delta method (applied to $U$ and $g(u) = 1/u$) yields

$$\mathbb{V}(\hat{\boldsymbol{q}}_t^\theta(y)) = g'(\mathbb{E}[U_1^2])\mathbb{V}(U) + o\left(\mathbb{V}(U)\right) = \frac{g'(\mathbb{E}[U_1^2])}{M}\mathbb{V}(U_1) + o\left(\mathbb{V}(U_1)/M\right).$$

In other words, $\mathbb{V}(\hat{\boldsymbol{q}}_t^\theta(y)) = \mathcal{O}\left(\frac{1}{M}\right)$. $\square$

## C.3 Proof of Theorem 3.2

**Exact reweighting identity.** Following Theorem 3.1, we can artificially introduce $\boldsymbol{q}_t^{\theta_{\text{old}}}(x)$ in the expression of $\nabla\ell_t(\theta)$, using $\nabla_\theta \boldsymbol{q}_t^\theta(x) = -\boldsymbol{q}_t^\theta(x)\sum_{\substack{y \in \mathcal{X} \\ y \neq x}} \boldsymbol{q}_t^\theta(y)\nabla_\theta \log r_{x,y}^t(\theta)$ and $\frac{\boldsymbol{q}_t^\theta(x)}{\boldsymbol{q}_t^{\theta_{\text{old}}}(x)} = \frac{\boldsymbol{q}_t^\theta(y)}{\boldsymbol{q}_t^{\theta_{\text{old}}}(y)}\frac{r_{x,y}^t(\theta_{\text{old}})}{r_{x,y}^t(\theta)}$. In fact, we obtain

$$\nabla\ell_t(\theta) = \sum_{x \in \mathcal{X}} \boldsymbol{q}_t^{\theta_{\text{old}}}(x)R(x)\sum_{\substack{y \in \mathcal{X} \\ y \neq x}} \boldsymbol{q}_t^\theta(y) \underbrace{\frac{\boldsymbol{q}_t^\theta(y)}{\boldsymbol{q}_t^{\theta_{\text{old}}}(y)}\frac{r_{x,y}^t(\theta_{\text{old}})}{r_{x,y}^t(\theta)}}_{=: \mathcal{W}_{xy}^\star(\theta)} \nabla_\theta \log r_{x,y}^t(\theta). \tag{13}$$

Note $\mathcal{W}_{xy}^\star(\theta) = \frac{\boldsymbol{q}_t^\theta(x)}{\boldsymbol{q}_t^{\theta_{\text{old}}}(x)}$ is independent of $y$, and by Assumption 3.2,

$$0 < \mathcal{W}_{xy}^\star(\theta) \leq \rho. \tag{14}$$

**Plug-in the estimator.** Define the two following quantities

$$\widehat{\mathcal{W}}_{xy}(\theta) := \frac{\boldsymbol{q}_t^\theta(y)}{\boldsymbol{q}_t^{\theta_{\text{old}}}(y)}\frac{s_{\theta_{\text{old}}}(x, T-t)_y}{s_\theta(x, T-t)_y}$$

$$\widehat{\nabla\ell_t}(\theta) := \sum_{x \in \mathcal{X}} \boldsymbol{q}_t^{\theta_{\text{old}}}(x)R(x)\sum_{\substack{y \in \mathcal{X} \\ y \neq x}} \boldsymbol{q}_t^\theta(y)\widehat{\mathcal{W}}_{xy}(\theta)\nabla_\theta \log s_\theta(x, T-t)_y.$$

By Assumption 3.2, $\frac{s_{\theta_{\text{old}}}/s_\theta}{r^t_{x,y}(\theta_{\text{old}})/r^t_{x,y}(\theta)} \in [e^{-2\delta}, e^{2\delta}]$, hence

$$\widehat{\mathcal{W}}_{xy}(\theta) \le e^{2\delta}\, \mathcal{W}^\star_{xy}(\theta) \le e^2\, \rho \quad \text{for } \delta \in (0,1]. \tag{15}$$

**Bias decomposition.** Subtract (13):

$$\widehat{\nabla \ell_t(\theta)} - \nabla \ell_t(\theta) = \sum_{x \in \mathcal{X}} \boldsymbol{q}^{\theta_{\text{old}}}(x) R(x) \sum_{\substack{y \in \mathcal{X} \\ y \ne x}} \boldsymbol{q}^\theta_t(y) \Big[ \underbrace{\widehat{\mathcal{W}}_{xy}\big(\nabla_\theta \log s_\theta - \nabla_\theta \log r^t_{x,y}(\theta)\big)}_{\text{(I)}}$$

$$+ \underbrace{\big(\widehat{\mathcal{W}}_{xy} - \mathcal{W}^\star_{xy}\big)\nabla_\theta \log r^t_{x,y}(\theta)}_{\text{(II)}} \Big].$$

*(I) Gradient replacement.* Using Assumption 3.2, (15), and $\sum_{y \ne x} \boldsymbol{q}^\theta_t(y) \le 1$,

$$\|\text{(I)}\| \le \delta \sum_x \boldsymbol{q}^{\theta_{\text{old}}}_t(x)|R(x)| \sum_{y \ne x} \boldsymbol{q}^\theta_t(y)\, \widehat{\mathcal{W}}_{xy}$$

$$\le \delta\, e^2 \rho \sum_x \boldsymbol{q}^{\theta_{\text{old}}}_t(x)|R(x)|$$

$$\le e^2 \rho\, R_{\max}\, \delta.$$

*(II) Ratio replacement.* From Assumption 3.2, $\left| \log \frac{s_{\theta_{\text{old}}}}{s_\theta} - \log \frac{r^t_{x,y}(\theta_{\text{old}})}{r^t_{x,y}(\theta)} \right| \le 2\delta \Rightarrow |\widehat{\mathcal{W}}_{xy} - \mathcal{W}^\star_{xy}| \le (e^{2\delta} - 1)\,\mathcal{W}^\star_{xy}$. Hence, using (14) and $\sum_{y \ne x} \boldsymbol{q}^\theta_t(y) \le 1$,

$$\|\text{(II)}\| \le (e^{2\delta} - 1) \sum_x \boldsymbol{q}^{\theta_{\text{old}}}_t(x)|R(x)| \sum_{y \ne x} \boldsymbol{q}^\theta_t(y)\, \mathcal{W}^\star_{xy}\, \|\nabla_\theta \log r^\theta_{xy}\|$$

$$\le (e^{2\delta} - 1)\, \rho\, (G + \delta) \sum_x \boldsymbol{q}^{\theta_{\text{old}}}_t(x)|R(x)|$$

$$\le (e^2 - 1)\, \rho\, (G + \delta)\, R_{\max}\, \delta,$$

where we used $e^{2\delta} - 1 \le (e^2 - 1)\delta$ for $\delta \in (0,1]$.

**Conclusion.** Combining (I) and (II),

$$\big\|\widehat{\nabla \ell_t(\theta)} - \nabla \ell_t(\theta)\big\| \le R_{\max} C(\rho, G)\delta.$$

where $C(\rho, G) = \rho(e^2 + (e^2 - 1)G) > 0$. The bound is independent of $\theta$, linear in $\delta$, and depends only on the trust-region radius $\rho$, the clipping level $G$, and $R_{\max}$.

### C.4 Proof of Proposition 3.4

To prove Proposition 3.4, we are going to prove the following explicit proposition regarding the linear system satisfied by $\nabla^\eta_\theta \boldsymbol{\pi}_\theta$.

**Proposition C.3.** *Let $\eta > 0$, and $D^\star_r$ denotes the block diagonal matrix defined in Lemma C.5. Then, $\nabla^\eta_\theta \boldsymbol{\pi}_\theta$ is the solution of the linear system*

$$A_\eta \boldsymbol{X} = B_\eta \in \mathbb{R}^{d \times p},$$

*where $A_\eta := D^\star_r [I_d - \eta\text{diag}(1/\boldsymbol{\pi}_\theta)] - I_d \in \mathbb{R}^{d \times d}$ and $B_\eta := -\eta D^\star_r \nabla_\theta \boldsymbol{\pi}_\theta / \boldsymbol{\pi}_\theta \in \mathbb{R}^{d \times p}$.*

The proof contains three parts and occupies the rest of this section:

1. recalling the implicit function theorem,
2. computing the matrices that appear in the linear system,
3. solving the linear system.

**Implicit function theorem.** Let us first recall the implicit function theorem.

> **Theorem C.4** (Implicit function theorem, Krantz and Parks 2002). *Let $U$ be an open subset of $\mathbb{R}^d \times \mathbb{R}^p$, and $f : U \to \mathbb{R}^d$ a continuously differentiable function. Let $(a, b) \in U$ such that $f(a, b) = 0$ and $\nabla_1 f(a, b)$ is invertible. Then, there exists an open set $W \subset \mathbb{R}^p$ containing $b$ and a function $g : W \to \mathbb{R}^d$ such that $g(b) = a$ and $\forall x \in W, f(g(x), x) = 0$. Moreover, $g$ is continuously differentiable and*
>
> $$\forall x \in W, \quad \nabla_1 f(a, b) \partial g(x) = -\nabla_2 f(a, b)$$
>
> *where $\partial g$ denotes the Jacobian of $g$ on $W$.*

A first point to note is that $\nabla_1 \mathcal{G}(\boldsymbol{p}, \theta) = \log(\boldsymbol{p}/\boldsymbol{\pi}_\theta) + \mathbf{1}$. Even if $\boldsymbol{\pi}_\theta = \operatorname*{argmin}_{\boldsymbol{p} \in \Delta_d} \mathcal{G}(\boldsymbol{p}, \theta)$, we have $\nabla_1 \mathcal{G}(\boldsymbol{\pi}_\theta, \theta) = 1 \neq 0$, because we compute the derivative in $\mathbb{R}^d$ and not in the probability simplex $\Delta_d$. This means that we cannot directly apply Theorem C.4 to $\nabla_1 \mathcal{G}(\boldsymbol{p}, \theta)$. To address this issue, we follow (Blondel et al., 2022) and consider $\mathcal{G}$ as a function of $\mathbb{R}^d \times \mathbb{R}^p$. Since we have a problem of the form $\boldsymbol{\pi}_\theta = \operatorname*{argmin}_{\boldsymbol{p} \in \Delta_d} \mathcal{G}(\boldsymbol{p}, \theta)$, we can define the fixed point operator

$$T_\eta(\boldsymbol{p}, \theta) = \operatorname{proj}_{\Delta_d}(\boldsymbol{p} - \eta \nabla_1 \mathcal{G}(\boldsymbol{p}, \theta)),$$

for $\eta > 0$. In fact, $T_\eta(\boldsymbol{\pi}_\theta, \theta) = \operatorname{proj}_{\Delta_d}(\boldsymbol{\pi}_\theta - \eta \mathbf{1})$ where $\operatorname{proj}_{\Delta_d} = \operatorname{sparsemax}$ (Martins and Astudillo, 2016; Rakotomandimby et al., 2024). From (Martins and Astudillo, 2016, Prop. 2), we have

$$\operatorname{sparsemax}(\boldsymbol{p} - \eta \mathbf{1}) = \operatorname{sparsemax}(\boldsymbol{p}), \quad \boldsymbol{p} \in \mathbb{R}^d.$$

This leads to $T_\eta(\boldsymbol{\pi}_\theta, \theta) = \boldsymbol{\pi}_\theta$, because $\operatorname{sparsemax}(\boldsymbol{\pi}_\theta) = \boldsymbol{\pi}_\theta$. We can therefore apply Theorem C.4 to the function $f_\eta(\boldsymbol{p}, \theta) = T_\eta(\boldsymbol{p}, \theta) - \boldsymbol{p}$.

**Computing the matrices.** Let $\eta > 0$, and define $h_\eta(\boldsymbol{p}, \theta) = \boldsymbol{p} - \eta \nabla_1 \mathcal{G}(\boldsymbol{p}, \theta)$. Then, we have $T_\eta(\boldsymbol{p}, \theta) = \operatorname{sparsemax}(h_\eta(\boldsymbol{p}, \theta))$ and $f_\eta(\boldsymbol{p}, \theta) = T_\eta(\boldsymbol{p}, \theta) - \boldsymbol{p}$. We note that since $\boldsymbol{\pi}_\theta \in \Delta_d$ and $T_\eta$ is a projection onto the probability simplex, $f_\eta(\boldsymbol{\pi}_\theta, \theta) = 0$. Following (Blondel et al., 2022, App. D) (this computation can be done by using the chain rule), we have

$$\begin{aligned}
\nabla_1 f_\eta(\boldsymbol{p}, \theta) &= D(\boldsymbol{p}, \theta)\left[I_d - \eta \nabla_{1,1} \mathcal{G}(\boldsymbol{p}, \theta)\right] - I_d \in \mathbb{R}^{d \times d}, \\
\nabla_2 f_\eta(\boldsymbol{p}, \theta) &= -\eta D(\boldsymbol{p}, \theta) \nabla_{1,2} \mathcal{G}(\boldsymbol{p}, \theta) \in \mathbb{R}^{d \times p},
\end{aligned} \tag{16}$$

where $D_r(\boldsymbol{p}, \theta) := \operatorname{diag}(r(h_\eta(\boldsymbol{p}, \theta))) - r(h_\eta(\boldsymbol{p}, \theta))r(h_\eta(\boldsymbol{p}, \theta))^{\mathrm{T}}/\|r(h_\eta(\boldsymbol{p}, \theta))\|_1 \in \mathbb{R}^{d \times d}$ and $r(h_\eta(\boldsymbol{p}, \theta)) \in \{0, 1\}^d$ (Martins and Astudillo, 2016). Here, for all $\boldsymbol{z} \in \mathbb{R}^d$, we define the vector $r(\boldsymbol{z})$ as follows:

$$\forall j \in [d], \ r(\boldsymbol{z})_j = \begin{cases} 1 & \text{if } \boldsymbol{z}_j > \tau(\boldsymbol{z}), \\ 0 & \text{otherwise,} \end{cases}$$

where $\tau$ is the unique function satisfying $\sum_{i=1}^d [\boldsymbol{z}_j - \tau(\boldsymbol{z})]_+ = 1$ for all $\boldsymbol{z} \in \mathbb{R}^d$. This definition is a bit tricky, but overall $r(h_\eta(\boldsymbol{p}, \theta))$ contains $k_h$ times the number 1 and $d - k_h$ times the number 0, where $k_h \in [d]$. This means that

$$D_r(\boldsymbol{p}, \theta) = \operatorname{diag}(r(h_\eta(\boldsymbol{p}, \theta))) - r(h_\eta(\boldsymbol{p}, \theta))r(h_\eta(\boldsymbol{p}, \theta))^{\mathrm{T}}/k_h.$$

We then obtain the following lemma.

> **Lemma C.5.** *Denote $\tilde{r}(\boldsymbol{z})$ as the vector with the sorted coordinates of $r(\boldsymbol{z})$, i.e., the vector with coordinates $\tilde{r}(\boldsymbol{z})_i = r(\boldsymbol{z})_{\sigma(i)}$, where $\sigma$ is the permutation such that $r(\boldsymbol{z})_{\sigma(1)} \geq \cdots \geq r(\boldsymbol{z})_{\sigma(d)}$. Then, $k_h = \max\{k \in [d] \mid 1 + k r(\boldsymbol{z})_{\sigma(k)} > \sum_{j=1}^k r(\boldsymbol{z})_{\sigma(j)}\}$ and,*
>
> $$D_{\tilde{r}}(\boldsymbol{p}, \theta) = R - \frac{1}{k_h} T,$$

where $R := \mathrm{diag}(\underbrace{1,\ldots,1}_{k_h\,\text{times}}, \underbrace{0,\ldots,0}_{d-k_h\,\text{times}})$ and $T := \begin{pmatrix} J & 0 \\ \hline 0 & 0 \end{pmatrix}$. Here, $J$ denotes the $k_h \times k_h$ matrix whose entries are all equal to $1$.

We can now replace the derivatives in Eq. (16) by their expression as $\nabla_{1,1}\mathcal{G}(\boldsymbol{p},\theta) = \mathrm{diag}(1/\boldsymbol{p})$ and $\nabla_{1,2}\mathcal{G}(\boldsymbol{p},\theta) = -\nabla_\theta\boldsymbol{\pi}_\theta/\boldsymbol{\pi}_\theta$. Following Theorem 3.1, we have

$$\nabla_1 f_\eta(\boldsymbol{p},\theta) = D(\boldsymbol{p},\theta)\left[I_d - \eta\,\mathrm{diag}(1/\boldsymbol{p})\right] - I_d, \quad \text{and} \quad \nabla_2 f_\eta(\boldsymbol{p},\theta) = \eta D(\boldsymbol{p},\theta)\left(\nabla_\theta\boldsymbol{\pi}_\theta/\boldsymbol{\pi}_\theta\right).$$

Let us define

$$D_r^\star := D_r(\boldsymbol{\pi}_\theta,\theta) = \mathrm{diag}(r(h_\eta(\boldsymbol{\pi}_\theta,\theta))) - r(h_\eta(\boldsymbol{\pi}_\theta,\theta))r(h_\eta(\boldsymbol{\pi}_\theta,\theta))^{\mathrm{T}}/k_h.$$

Since $h_\eta(\boldsymbol{\pi}_\theta,\theta)) = \boldsymbol{\pi}_\theta - \eta\mathbf{1}$, we find from Lemma C.5 that if $\eta \le \min_{i\in[d]}\boldsymbol{\pi}_i(\theta)$, then $k_h = d$.

Let us continue the proof without additional assumptions on $\eta$. We begin by verifying that the matrix

$$-A_{\tilde{r}}^\star := \nabla_1 f_\eta(\boldsymbol{\pi}_\theta,\theta) = D_r^\star\left[I_d - \eta\,\mathrm{diag}(1/\boldsymbol{\pi}_\theta)\right] - I_d$$

is invertible. Without loss of generality, instead of reordering the elements of the canonical basis with $\sigma$ (Lemma C.5), we can check the invertibility directly on $A_{\tilde{r}}^\star$ as follows.

$$-A_{\tilde{r}}^\star = \left(R - \frac{1}{k_h}T\right)\left[I_d - \eta\,\mathrm{diag}(1/\boldsymbol{\pi}_\theta)\right] - I_d$$

$$= R - \eta R\,\mathrm{diag}(1/\boldsymbol{\pi}_\theta) - \frac{1}{k_h}T + -\frac{\eta}{k_h}T\,\mathrm{diag}(1/\boldsymbol{\pi}_\theta) - I_d$$

$$= \mathrm{diag}(\underbrace{\frac{\eta}{\boldsymbol{\pi}_\theta(\sigma(1))},\ldots,\frac{\eta}{\boldsymbol{\pi}_\theta(\sigma(k_h))}}_{k_h\,\text{times}}, \underbrace{1,\ldots,1}_{d-k_h\,\text{times}}) + \frac{1}{k_h}\mathbf{1}_d^{(k_h)}\left(\mathbf{1}_d^{(k_h)} - \eta w(\theta)^{(k_h)}\right)^{\mathrm{T}},$$

where the second inequality is due to Lemma C.5, and

$$\mathbf{1}_d^{(k_h)} := (\underbrace{1,\ldots,1}_{k_h\,\text{times}}, \underbrace{0,\ldots,0}_{d-k_h\,\text{times}})^{\mathrm{T}}, \quad w(\theta)^{(k_h)} := (\underbrace{\frac{1}{\boldsymbol{\pi}_\theta(\sigma(1))},\ldots,\frac{1}{\boldsymbol{\pi}_{\sigma(d)}(\theta)}}_{k_h\,\text{times}}, \underbrace{0,\ldots,0}_{d-k_h\,\text{times}})^{\mathrm{T}}.$$

We then find that $-A_{\tilde{r}}^\star$ is a rank-one update of a diagonal matrix, which can be inverted explicitly using the Sherman–Morrison formula (Bartlett, 1951).

**Lemma C.6** (Sherman–Morrison formula, Bartlett 1951)**.** *Let $M \in \mathbb{R}^{d\times d}$ be an invertible matrix and $u,v \in \mathbb{R}^d$. Then, $M + uv^{\mathrm{T}}$ is invertible if and only if $1 + v^{\mathrm{T}}M^{-1}u \ne 0$, and*

$$(M + uv^{\mathrm{T}})^{-1} = M^{-1} + \frac{M^{-1}uv^{\mathrm{T}}M^{-1}}{1 + v^{\mathrm{T}}M^{-1}u}.$$

We replace the elements in Lemma C.6 by our variables as $M = \mathrm{diag}(\eta/\boldsymbol{\pi}_\theta(\sigma(1)),\ldots,\eta/\boldsymbol{\pi}_\theta(\sigma(k_h)),1,\ldots,1)$ is invertible, $u = \frac{1}{k_h}\mathbf{1}_d^{(k_h)}$ and $v = \mathbf{1}_d^{(k_h)} - \eta w(\theta)^{(k_h)}$. Then,

$$1 + v^{\mathrm{T}}M^{-1}u = 1 + \sum_{i=1}^{k_h}\frac{1 - \eta/\boldsymbol{\pi}_\theta(\sigma(i))}{k_h\eta/\boldsymbol{\pi}_\theta(\sigma(i))} = \frac{1}{k_h\eta}\sum_{i=1}^{k_h}\boldsymbol{\pi}_\theta(\sigma(i)) > 0.$$

The nonzero terms in the numerator are located in the upper-left square of size $k_h$, and for $1 \le i,j \le k_h$,

$$[M^{-1}uv^{\mathrm{T}}M^{-1}]_{i,j} = \frac{\boldsymbol{\pi}_\theta(\sigma(i))}{\eta}\frac{1}{k_h}\left(1 - \frac{\eta}{\boldsymbol{\pi}_\theta(\sigma(j))}\right)\frac{\boldsymbol{\pi}_\theta(\sigma(j))}{\eta}$$

$$= \frac{\boldsymbol{\pi}_\theta(\sigma(i))\boldsymbol{\pi}_\theta(\sigma(j))}{k_h\eta^2}\left(1 - \frac{\eta}{\boldsymbol{\pi}_\theta(\sigma(j))}\right)$$

$$= \frac{\boldsymbol{\pi}_\theta(\sigma(i))}{k_h\eta}\left(\frac{\boldsymbol{\pi}_\theta(\sigma(j))}{\eta} - 1\right).$$

This shows that $-A_r^\star$ is invertible, and its inverse is given as

$$
-A_r^{\star-1} = \mathrm{diag}\left( \underbrace{\frac{\boldsymbol{\pi}_\theta(\sigma(1))}{\eta}, \dots, \frac{\boldsymbol{\pi}_\theta(\sigma(k_h))}{\eta}}_{k_h\,\text{times}}, \underbrace{1, \dots, 1}_{d-k_h\,\text{times}} \right) + L,
$$

where $L$ is the rank-one matrix defined as

$$
L_{ij} = \begin{cases} \dfrac{\boldsymbol{\pi}_\theta(\sigma(i))}{\sum_{l=1}^{k_h} \boldsymbol{\pi}_{\sigma(l)}(\theta)} \left( \dfrac{\boldsymbol{\pi}_\theta(\sigma(j))}{\eta} - 1 \right) & \text{if } 1 \le i, j \le k_h, \\ 0 & \text{otherwise.} \end{cases}
$$

We can now apply Theorem C.4 to conclude that $\nabla_\theta^\eta \boldsymbol{\pi}_\theta$ is the solution to the linear system

$$
A_\eta \boldsymbol{X} = B_\eta \in \mathbb{R}^{d \times p},
$$

where $A_\eta := D_r^\star [I_d - \eta \mathrm{diag}(1/\boldsymbol{\pi}_\theta)] - I_d \in \mathbb{R}^{d \times d}$ and $B_\eta := -\eta D_r^\star \nabla_\theta \boldsymbol{\pi}_\theta / \boldsymbol{\pi}_\theta \in \mathbb{R}^{d \times p}$.

**Solving the linear system** The matrix product gives us that

$$
\nabla_\theta^\eta \boldsymbol{\pi}_\theta = S \nabla_\theta \boldsymbol{\pi}_\theta
$$

where $S := z \mathbf{1}^{\mathrm{T}}$ with $z_i = \begin{cases} \dfrac{k_h \boldsymbol{\pi}_\theta(\sigma(i))}{\sum_{l=1}^{k_h} \boldsymbol{\pi}_{\sigma(l)}(\theta)} & \text{if } 1 \le i \le k_h, \\ 0 & \text{otherwise.} \end{cases}$

$\nabla_\theta^\eta \boldsymbol{\pi}_\theta$ is then a weighted version of $\nabla_\theta \boldsymbol{\pi}_\theta$. In fact, $\nabla_\theta^\eta \boldsymbol{\pi}_\theta = z \cdot (\nabla_{\theta_1} \boldsymbol{\pi}_\theta, \dots, \nabla_{\theta_p} \boldsymbol{\pi}_\theta)$ each gradient is then weighted by $z_i$.

Interestingly, as in (Blondel et al., 2022), $\eta$ does not appear directly in $S$. In our case, the dependence is implicit through $k_h$. We will then choose $\eta$ so that $k_h$ corresponds to theo the term we have in our Monte Carlo approximation of the policy gradient. By doing this, we will be able to totally compute $z$ since we have access to $\boldsymbol{\pi}_{\sigma(i)}$, $\forall 1 \le i \le k_h$ through the concrete score $s_\theta(\sigma(i), 0)$.

## C.5 Proof of Lemma 3.3

The proof is a direct consequence of (Campbell et al., 2022, Prop. 4) and (Maas, 2011, Thm. 4.7).

**Proposition C.7** (Campbell et al. 2022)**.** *The corrector rate matrix $Q_t^c := Q_t + \overline{Q}_t$ has $\boldsymbol{p}_t$ as its stationary distribution.*

Then, the rate matrix $Q_t^c$ has $\boldsymbol{p}_t$ as a stationary distribution. To provide a gradient flow interpretation, Maas 2011 introduces a specific Wasserstein-like metric $\mathcal{W} : \Delta_d \times \Delta_d \to \mathbb{R}$. We refer to (Maas, 2011, Sec. 1) for a complete definition.

**Remark C.1.** *This metric is really close to the Wassertein metric, as it has a transport-cost interpretation. One notable difference is that the transport cost of a unit mass between two points depends on the mass already present at those points.*

We state one of the main results of (Maas, 2011).

**Theorem C.8.** *[Maas 2011] Let $\Delta$ be a rate matrix of stationary distribution $\nu$. Then, $\frac{\mathrm{d}\boldsymbol{p}_t}{\mathrm{d}t} = \Delta \boldsymbol{p}_t$ is a gradient flow trajectory for the functional $\mathcal{H}(\boldsymbol{p}) = \mathrm{KL}(\boldsymbol{p}\|\nu)$ with respect to $\mathcal{W}$.*

Combining these two results means that sampling from the ODE

$$
\frac{\mathrm{d}\boldsymbol{p}_t}{\mathrm{d}t} = Q_t^c \boldsymbol{p}_t, \quad \text{where } Q_t^c := Q_t + \overline{Q}_t
$$

implements a gradient flow for $\mathrm{KL}(\,\cdot\,\|\boldsymbol{p}_t)$ in $\Delta_d$, with respect to $\mathcal{W}$.

## C.6 Proof of Theorem 3.5

The proof contains two parts and occupies the rest of this section:

1. proving technical lemmas,

2. main proof.

### C.6.1 Technical lemmas

We first recall the expression of the functional derivative of the KL over $\Delta_d$ for our setup.

**Lemma C.9** (Functional derivatives). *For KL divergence, the chain rule for functional derivatives can be written as*

$$\frac{\mathrm{d}\,\mathrm{KL}(\boldsymbol{q}_s \,||\, \boldsymbol{\pi}(\theta_s))}{\mathrm{d}s} = \sum_{x \in \mathcal{X}} \frac{\delta\,\mathrm{KL}(\boldsymbol{q}_s \,||\, \boldsymbol{\pi}(\theta_s))}{\delta \boldsymbol{q}_s} \frac{\partial \boldsymbol{q}_s}{\partial s} + \sum_{x \in \mathcal{X}} \frac{\delta\,\mathrm{KL}(\boldsymbol{q}_s \,||\, \boldsymbol{\pi}(\theta_s))}{\delta \boldsymbol{\pi}(\theta_s)} \frac{\partial \boldsymbol{\pi}(\theta_s)}{\partial s}.$$

We provide a useful bound on $\|\nabla_\theta \log \boldsymbol{\pi}_{\theta_s}\|$.

**Lemma C.10** (Bound on the derivative of the log distribution). *Under Assumption 3.4, we have that*
$$\|\nabla_\theta \log \boldsymbol{\pi}(\theta)\| \le C_{\log}, \quad \text{for } \theta \in \mathbb{R}^p$$
*where $C_{\log} > 0$.*

*Proof.* $\nabla_\theta \log \boldsymbol{\pi}(\theta) = \frac{\nabla_\theta \boldsymbol{\pi}(\theta)}{\boldsymbol{\pi}(\theta)}$. Then, from Assumption 3.4, for $\theta \in \mathbb{R}^p$, $\|\nabla_\theta \log \boldsymbol{\pi}(\theta)\| \le C_{\log} :=$ $\frac{C}{\varepsilon}$.

$\square$

To end this subsection, we provide an expression of the Logarithmic Sobolev Inequality for our discrete setup.

**Lemma C.11** (Log-Sobolev inequality, Diaconis and Saloff-Coste 1996).
$$\mathrm{KL}(\boldsymbol{q}_s \| \boldsymbol{\pi}) \le \frac{1}{2\mu} \sum_{x,y \in \mathcal{X}} Q_{T-T_0}^{c,\theta_s}(x,y) \boldsymbol{q}_s(y) \left( \log \frac{\boldsymbol{q}_s(y)}{\boldsymbol{\pi}_y(\theta_s)} - \log \frac{\boldsymbol{q}_s(x)}{\boldsymbol{\pi}_x(\theta_s)} \right) \quad \text{for } \mu > 0.$$

*Proof.* Thanks to Theorem C.8, we can apply the Log-Sobolev Inequality (LSI) of (Diaconis and Saloff-Coste, 1996):

$$\mathrm{Ent}_{\boldsymbol{\pi}(\theta_s)}(f^2) \le \frac{1}{2\mu} \sum_{x,y} Q_{T-T_0}^{c,\theta_s}(x,y) \boldsymbol{\pi}_y(\theta_s)(f(y) - f(x)) \log \left( \frac{f(y)}{f(x)} \right),$$

to $f(x) = \sqrt{\frac{\boldsymbol{q}_s(y)}{\boldsymbol{\pi}_y(\theta_s)}}$.

**Remark C.2.** *We applied (Diaconis and Saloff-Coste, 1996, Lem. 2.7) to the standard LSI form in the paper.*

$\square$

### C.6.2 Main proof

We recall the coupled equations stated in the main document:

$$\frac{\mathrm{d}\boldsymbol{q}_s}{\mathrm{d}s} = Q_{T-T_0}^{c,\theta_s}\boldsymbol{q}_s, \tag{17}$$

$$\frac{\mathrm{d}\theta_s}{\mathrm{d}s} = -\beta_s\Gamma(\boldsymbol{q}_s,\theta_s). \tag{18}$$

We provide below the main proof of Theorem 3.5, inspired by (Marion et al., 2024).

**Evolution of the loss.**  Since $\nabla\ell^A(\theta_s) = \Gamma(\boldsymbol{\pi}_{\theta_s},\theta_s)$, we have,

$$\begin{aligned}
\frac{\mathrm{d}\ell^A}{\mathrm{d}s}(s) &= \left\langle \nabla\ell^A(\theta_s), \frac{\mathrm{d}}{\mathrm{d}s}\theta_s \right\rangle = -\beta_s\left\langle \nabla\ell^A(\theta_s), \Gamma(\boldsymbol{q}_s,\theta_s)\right\rangle \\
&= -\beta_s\left\langle \nabla\ell^A(\theta_s), \Gamma(\boldsymbol{\pi}_{\theta_s},\theta_s)\right\rangle + \beta_s\left\langle \nabla\ell^A(\theta_s), \Gamma(\boldsymbol{\pi}_{\theta_s},\theta_s) - \Gamma(\boldsymbol{q}_s,\theta_s)\right\rangle \\
&\leq -\beta_s\left\|\nabla\ell^A(\theta_s)\right\|^2 + \beta_s\left\|\nabla\ell^A(\theta_s)\right\|\left\|\Gamma(\boldsymbol{\pi}_{\theta_s},\theta_s) - \Gamma(\boldsymbol{q}_s,\theta_s)\right\|.
\end{aligned}$$

Then, by Assumption 3.5,

$$\begin{aligned}
\frac{\mathrm{d}\ell^A}{\mathrm{d}s}(s) &\leq -\beta_s\left\|\nabla\ell^A(\theta_s)\right\|^2 + \beta_s C_\Gamma\left\|\nabla\ell^A(\theta_s)\right\|\sqrt{\mathrm{KL}(\boldsymbol{q}_s\,\|\,\boldsymbol{\pi}_{\theta_s})} \\
&\leq -\frac{1}{2}\beta_s\left\|\nabla\ell^A(\theta_s)\right\|^2 + \frac{1}{2}\beta_s C_\Gamma^2\,\mathrm{KL}(\boldsymbol{q}_s\,\|\,\boldsymbol{\pi}_{\theta_s}),
\end{aligned}$$

where we used the inequality $ab \leq \frac{1}{2}(a^2+b^2)$.

**Evolution of the KL divergence of $\boldsymbol{q}_s$ from $\boldsymbol{\pi}_{\theta_s}$.**  Since $\mathrm{KL}(\boldsymbol{q}_s\,\|\,\boldsymbol{\pi}_{\theta_s}) = \sum_{x\in\mathcal{X}}\log\left(\frac{\boldsymbol{q}_s}{\boldsymbol{\pi}_{\theta_s}}\right)\boldsymbol{q}_s$, we have that

$$\frac{\delta\,\mathrm{KL}(\boldsymbol{q}_s\,\|\,\boldsymbol{\pi}_{\theta_s})}{\delta\boldsymbol{q}_s} = \log\left(\frac{\boldsymbol{q}_s}{\boldsymbol{\pi}_{\theta_s}}\right) + 1, \quad\text{and}\quad \frac{\delta\,\mathrm{KL}(\boldsymbol{q}_s\,\|\,\boldsymbol{\pi}_{\theta_s})}{\delta\boldsymbol{\pi}_{\theta_s}} = -\frac{\boldsymbol{q}_s}{\boldsymbol{\pi}_{\theta_s}}.$$

Lemma C.9 gives us that

$$\frac{\mathrm{d}\,\mathrm{KL}(\boldsymbol{q}_s\,\|\,\boldsymbol{\pi}_{\theta_s})}{\mathrm{d}s} = \underbrace{\sum_{x\in\mathcal{X}}\log\left(\frac{\boldsymbol{q}_s}{\boldsymbol{\pi}_{\theta_s}}\right)\frac{\mathrm{d}\boldsymbol{q}_s}{\mathrm{d}s}}_{a} - \underbrace{\sum_{x\in\mathcal{X}}\frac{\boldsymbol{q}_s}{\boldsymbol{\pi}_{\theta_s}}\frac{\partial\boldsymbol{\pi}_{\theta_s}}{\partial s}}_{b}, \tag{19}$$

since $\sum\frac{\mathrm{d}\boldsymbol{q}_s}{\mathrm{d}s} = \frac{\mathrm{d}}{\mathrm{d}s}\sum\boldsymbol{q}_s = \frac{\mathrm{d}}{\mathrm{d}s}1 = 0$.

**Analysis of $a$.**  First, from Eq. (17), $\frac{\mathrm{d}\boldsymbol{q}_s}{\mathrm{d}s}(x) = (Q_{T-T_0}^{c,\theta_s}\boldsymbol{q}_s)(x) = \sum_y Q_{T-T_0}^{c,\theta_s}(x,y)\boldsymbol{q}_s(y)$. Then, since $Q_{T-T_0}^{c,\theta_s}$ sum to zero, we can write that

$$\begin{aligned}
a &= \sum_{x,y}\log\left(\frac{\boldsymbol{q}_s(x)}{\boldsymbol{\pi}_{\theta_s}(x)}\right)Q_{T-T_0}^{c,\theta_s}(x,y)\boldsymbol{q}_s(y) \\
&= \frac{1}{2}\sum_{x,y}\left(\log\left(\frac{\boldsymbol{q}_s(x)}{\boldsymbol{\pi}_{\theta_s}(x)}\right) - \log\left(\frac{\boldsymbol{q}_s(y)}{\boldsymbol{\pi}_{\theta_s}(y)}\right)\right)Q_{T-T_0}^{c,\theta_s}(x,y)\boldsymbol{q}_s(y).
\end{aligned}$$

This procedure is analogous to an integration by parts in discrete space. We can then apply Lemma C.11,

$$a \leq -2\mu\,\mathrm{KL}(\boldsymbol{q}_s\,\|\,\boldsymbol{\pi}_{\theta_s}).$$

**Analysis of $b$.**  By the chain rule, $\frac{\partial\boldsymbol{\pi}_{\theta_s}}{\partial s} = \left\langle\nabla_\theta\boldsymbol{\pi}_{\theta_s}, \frac{\mathrm{d}}{\mathrm{d}s}\theta_s\right\rangle$. By plugging in Eq. (17), we can rewrite $b$ as

$$b = -\beta_s\left\langle\Psi(\boldsymbol{q}_s,\theta_s),\Gamma(\boldsymbol{q}_s,\theta_s)\right\rangle,$$

where $\Psi(\boldsymbol{q}_s, \theta_s) = \int \boldsymbol{q}_s \nabla_\theta \log \boldsymbol{\pi}_{\theta_s}$. Then,

$$\|\Psi(\boldsymbol{q}_s, \theta_s) - \Psi(\boldsymbol{\pi}_{\theta_s}, \theta_s)\| = \left\|\int (\boldsymbol{q}_s - \boldsymbol{\pi}_{\theta_s}) \nabla_\theta \log \boldsymbol{\pi}_{\theta_s}\right\| \leq C_{\log} 2 \|\boldsymbol{q}_s - \boldsymbol{\pi}_{\theta_s}\|_{\mathrm{TV}}$$

$$\leq C_{\log} \sqrt{2 \operatorname{KL}(\boldsymbol{q}_s \| \boldsymbol{\pi}_{\theta_s})}.$$

where we used Lemma C.10 for the first inequality and Pinsker's inequality for the second. Note that $\Psi(\boldsymbol{\pi}_{\theta_s}, \theta_s) = \sum \boldsymbol{\pi}_{\theta_s} \nabla_\theta \log \boldsymbol{\pi}_{\theta_s} = \sum \nabla_\theta \boldsymbol{\pi}_{\theta_s} = \nabla_\theta \sum \boldsymbol{\pi}_{\theta_s} = \nabla_\theta 1 = 0$. By Cauchy-Schwarz inequality, we have that $|b| \leq \beta_s K \sqrt{\operatorname{KL}(\boldsymbol{q}_s \| \boldsymbol{\pi}_{\theta_s})}$, where $K = C_{\log} C_\Gamma \sqrt{2}$.

**Bounding the KL divergence of $q_s$ from $\pi_{\theta_s}$.** Combining the bounds on $a$ and $b$ with Eq. (19) yields

$$\frac{\mathrm{d}}{\mathrm{d}s} \operatorname{KL}(\boldsymbol{q}_s \| \boldsymbol{\pi}_{\theta_s}) \leq -2\mu \operatorname{KL}(p_t \| \boldsymbol{\pi}_{\theta_s}) + \beta_s K \sqrt{\operatorname{KL}(\boldsymbol{q}_s \| \boldsymbol{\pi}_{\theta_s})}.$$

Let $y(s) = \operatorname{KL}(\boldsymbol{q}_s \| \boldsymbol{\pi}_{\theta_s})$. We can rewrite

$$\frac{\mathrm{d}}{\mathrm{d}s} y(s) \leq -2\mu y(s) + \beta_s K \sqrt{y(s)}.$$

Then, by writing $\frac{\mathrm{d}}{\mathrm{d}s} y = 2\sqrt{y} \frac{\mathrm{d}}{\mathrm{d}s} \sqrt{y}$, we have $\frac{\mathrm{d}}{\mathrm{d}s} \sqrt{y} \leq -\mu \sqrt{y(s)} + \frac{\beta_s K}{2}$. Let us introduce $\Phi(s) := e^{\mu s} u(s)$ where $u(s) = \sqrt{y(s)}$ such that

$$\Phi'(s) = e^{\mu s}(u'(s) + \mu u(s)) \leq e^{\mu s} \frac{\beta_s K}{2}.$$

Let $s \geq 0$, integrating this inequality over $[0, s]$ yields

$$\Phi(s) \leq \Phi(0) + \int_0^s e^{\mu t} \frac{\beta_t K}{2} \, \mathrm{d}t.$$

Hence,

$$\sqrt{y(s)} \leq \sqrt{y(0)} e^{-\mu s} + \frac{K}{2} e^{-\mu s} \int_0^s \beta_t e^{\mu t} \, \mathrm{d}t. \tag{20}$$

**Bounding the loss function.** We recall the bound on the loss:

$$\frac{\mathrm{d}\ell^A}{\mathrm{d}s}(s) \leq -\frac{1}{2} \beta_s \left\|\nabla \ell^A(\theta_s)\right\|^2 + \frac{1}{2} \beta_s C_\Gamma^2 y(s).$$

By integrating between $0$ and $S$, and exploiting the fact that we can bound $\beta_s$ by $\beta_S$ since $\beta_s$ is decreasing, we have

$$\frac{1}{S} \int_0^S \left\|\nabla \ell^A(\theta_s)\right\|^2 \mathrm{d}s \leq \frac{2}{S\beta_S}(\ell^A(0) - \inf \ell^A) + \frac{C_\Gamma^2}{S\beta_S} \int_0^S \beta_s y(s) \, \mathrm{d}s. \tag{21}$$

Recall that, by assumption of the Theorem, $\beta_s = \min(1, \frac{1}{\sqrt{s}})$. Thus $S\beta_S = \sqrt{S}$.

**Analysis of the last integral** From Eq. (20), we can bound $\int_0^S \beta_s y(s) \mathrm{d}s$. In fact,

$$\int_0^S \beta_s y(s) \mathrm{d}s \leq y(0) \int_0^S \beta_s e^{-2\mu s} \mathrm{d}s + \sqrt{y(0)} K \int_0^S \beta_s e^{-2\mu s} \left(\int_0^s \beta_u e^{\mu u} \mathrm{d}u\right) \mathrm{d}s$$

$$+ \frac{K^2}{4} \int_0^S \beta_s e^{-2\mu s} \left(\int_0^s \beta_u e^{\mu u} \mathrm{d}u\right)^2 \mathrm{d}s$$

$$\leq \frac{y(0)\beta_0}{2\mu} \left(1 - e^{-2\mu S}\right) + \sqrt{y(0)} \frac{K\beta_0^2}{\mu^2} \left(\frac{1}{2} - e^{-\mu S} + \frac{1}{2} e^{-2\mu S}\right)$$

$$+ \frac{K^2}{4} \int_0^S \beta_s \left(\int_0^s \beta_u e^{\mu(u-s)} \mathrm{d}u\right)^2 \mathrm{d}s.$$

The first two terms are converging when $S \to \infty$. Let us analyze the last term by defining $I(s) = \int_0^s \beta_u e^{\mu(u-s)} du$. Let $S_0 \geq 2$ (depending only on $\mu$) such that $\frac{\ln(S_0)}{\mu} \leq \frac{S_0}{2}$. For $s \geq S_0$, let $\alpha(s) := s - \frac{\ln s}{\mu}$. We have, for $s \geq S_0$,

$$I(s) = \int_0^{\alpha(s)} \beta_u e^{\mu(u-s)} du + \int_{\alpha(s)}^s \beta_u e^{\mu(u-s)} du \leq \beta_0 e^{-\mu s} \int_0^{\alpha(s)} e^{\mu u} du + (s - \alpha(s))\beta_{\alpha(s)}$$

$$\leq \frac{\beta_0}{\mu} e^{\mu(\alpha(s)-s)} + \frac{\beta_{\alpha(s)} \ln s}{\mu} \leq \frac{\beta_0}{\mu s} + \frac{\beta_{s/2} \ln s}{\mu},$$

where in the last inequality we used that $\alpha(s) \geq s/2$ and $\beta_s$ is decreasing. This means that for $s \geq T_0$,

$$(I(s))^2 \leq \frac{\beta_0^2}{\mu^2 s^2} + 2\frac{\beta_0 \beta_{s/2}}{\mu^2} \frac{\ln s}{s} + \frac{\beta_{s/2}^2 (\ln s)^2}{\mu^2}.$$

For $s < S_0$, we can simply bound $I(s)$ by $\beta_0 S_0$. We obtain

$$\int_0^S \beta_s (I(s))^2 ds \leq \int_0^{S_0} \beta_s \beta_0^2 S_0^2 ds$$

$$+ \frac{1}{\mu^2} \left( \int_{S_0}^S \beta_0^2 \frac{\beta_s}{s^2} ds + \int_{S_0}^S 2\beta_0 \frac{\beta_s \beta_{s/2} \ln s}{s} ds + \int_{S_0}^S \beta_s \beta_{s/2}^2 (\ln s)^2 ds \right).$$

Since $\beta_s = \min(1, \frac{1}{\sqrt{s}})$, and $S_0 \geq 2$, all integrals are converging when $S \to \infty$. Plugging this into (21), we finally obtain the existence of a constant $c > 0$ such that

$$\frac{1}{S} \int_0^S \|\nabla \ell^A(\theta_s)\|^2 ds \leq \frac{c}{S^{1/2}}.$$

## D  Mathematical supplements

### D.1  First Variation in the discrete setup on $\Delta_d$

**Overview**  In the discrete setup, when considering a functional $\mathcal{F}(\boldsymbol{p})$ defined on the probability simplex $\Delta_d$, the first variation quantifies the sensitivity of $\mathcal{F}$ to perturbations in the probability distribution $\boldsymbol{p} = \{p_1, \ldots, p_d\}$. For a small perturbation $\boldsymbol{p} \to \boldsymbol{p} + \epsilon \eta$, the first variation is given by

$$\delta\mathcal{F}(\boldsymbol{p}; \eta) = \lim_{\epsilon \to 0} \frac{\mathcal{F}(\boldsymbol{p} + \epsilon\eta) - \mathcal{F}(\boldsymbol{p})}{\epsilon}.$$

In practice, the first variation can often be expressed as a weighted sum over the components of $\eta$, as

$$\delta\mathcal{F}(\boldsymbol{p}; \eta) = \sum_{i=1}^d \frac{\partial \mathcal{F}}{\partial p_i} \eta_i,$$

where $\frac{\partial \mathcal{F}}{\partial p_i}$ denotes the partial derivative of $\mathcal{F}$ with respect to $p_i$, which is the quantity of interest. The next two paragraphs detail the derivation for the two functionals considered in this paper.

**First Variation of $\mathcal{F}(\boldsymbol{p}) = \mathbb{E}_{x \sim \boldsymbol{p}}[R(x)]$.**  The functional can be written as:

$$\mathcal{F}(\boldsymbol{p}) = \sum_{i=1}^d p_i R(x_i),$$

where $\boldsymbol{p} = \{p_1, \ldots, p_d\}$ is a probability vector, and $R(x_i)$ represents the value of $R$ at $x_i$. Consider a small perturbation $\boldsymbol{p} \to \boldsymbol{p} + \epsilon\eta$. Then, $\mathcal{F}(\boldsymbol{p} + \epsilon\eta) = \sum_{i=1}^d (p_i + \epsilon\eta_i)R(x_i)$. After expanding this first order in $\epsilon$, we obtain

$$\mathcal{F}(\boldsymbol{p} + \epsilon\eta) = \sum_{i=1}^d p_i R(x_i) + \epsilon \sum_{i=1}^d \eta_i R(x_i) + o(\epsilon).$$

Thus, the first variation is

$$\delta\mathcal{F}(\boldsymbol{p};\eta) = \lim_{\epsilon\to 0}\frac{\mathcal{F}(\boldsymbol{p}+\epsilon\eta)-\mathcal{F}(\boldsymbol{p})}{\epsilon} = \sum_{i=1}^{d}\eta_i R(x_i),$$

which leads to

$$\mathbf{f} = \left[\frac{\partial\mathcal{F}}{\partial p_i}\right]_{1\le i\le d} = [R(x_i)]_{1\le i\le d}.$$

**First Variation of** $\mathcal{F}(\boldsymbol{p}) = \mathrm{KL}(\boldsymbol{p}\|\mathbf{q}).$ Consider the small perturbation of the functional as $\mathcal{F}(\boldsymbol{p}+\epsilon\eta) = \sum_{i=1}^{n}(p_i+\epsilon\eta_i)\ln\frac{p_i+\epsilon\eta_i}{q_i}$, and its expansion to the first order in $\epsilon$:

$$\mathcal{F}(\boldsymbol{p}+\epsilon\eta) = \sum_{i=1}^{d}(p_i+\epsilon\eta_i)\left(\ln\frac{p_i}{q_i}+\frac{\epsilon\eta_i}{p_i}\right)+o(\epsilon).$$

Keeping only the terms linear in $\epsilon$, we find that

$$\mathcal{F}(\boldsymbol{p}+\epsilon\eta) = \sum_{i=1}^{d}p_i\ln\frac{p_i}{q_i}+\epsilon\sum_{i=1}^{d}\eta_i\ln\frac{p_i}{q_i}+\epsilon\sum_{i=1}^{d}\eta_i+o(\epsilon).$$

Therefore,

$$\delta\mathcal{F}(\boldsymbol{p};\eta) = \lim_{\epsilon\to 0}\frac{\mathcal{F}(\boldsymbol{p}+\epsilon\eta)-\mathcal{F}(\boldsymbol{p})}{\epsilon} = \sum_{i=1}^{d}\eta_i\left(\ln\frac{p_i}{q_i}+1\right).$$

which leads us to

$$\mathbf{f} = \left[\frac{\partial\mathcal{F}}{\partial p_i}\right]_{1\le i\le d} = \left[\ln\frac{p_i}{q_i}+1\right]_{1\le i\le d}.$$

## D.2 KL regularization term

For $t\in[0,T]$, we take advantage of the computation of (Zhang et al., 2025, Lem. 1) to $\boldsymbol{q}_t^{\theta}$ and $\boldsymbol{q}_t^{\theta_{\mathrm{pre}}}$. In fact, $\mathrm{KL}(\boldsymbol{q}_t^{\theta}\|\boldsymbol{q}_t^{\theta_{\mathrm{pre}}}) = \mathbb{E}_{x_t\sim\boldsymbol{q}_t^{\theta}}\left[\frac{\mathrm{d}\boldsymbol{q}_t^{\theta}}{\mathrm{d}\boldsymbol{q}_t^{\theta_{\mathrm{pre}}}}\right]$. By applying Girsanov's theorem (Palmowski and Rolski, 2002) to the Radon-Nikodym derivative as in (Zhang et al., 2025), a generalized $I$-divergence term appears (Amari, 2012). We obtain

$$\mathrm{KL}(\boldsymbol{q}_t^{\theta}\|\boldsymbol{q}_t^{\theta_{\mathrm{pre}}}) = \mathbb{E}_{x_t\sim\boldsymbol{q}_t^{\theta}}\left[\sum_{\substack{y\in\mathcal{X}\\y\ne x}}\overline{Q}_t^{\theta_{\mathrm{pre}}}(x_t,y)-\overline{Q}_t^{\theta}(x_t,y)+\overline{Q}_t^{\theta}(x_t,y)\log\frac{\overline{Q}_t^{\theta}(x_t,y)}{\overline{Q}_t^{\theta_{\mathrm{pre}}}(x_t,y)}\right].$$

Once integrated on the whole path from $t=0$ to $t=T$, we recover the result from (Wang et al., 2025):

$$\mathrm{KL}(\boldsymbol{q}_{[0,T]}^{\theta}\|\boldsymbol{q}_{[0,T]}^{\theta_{\mathrm{pre}}}) =$$

$$\int_0^T\mathbb{E}_{x_{[0,T]}\sim\boldsymbol{q}_{[0,T]}^{\theta}}\left[\sum_{\substack{y\in\mathcal{X}\\y\ne x}}\overline{Q}_t^{\theta_{\mathrm{pre}}}(x_t,y)-\overline{Q}_t^{\theta}(x_t,y)+\overline{Q}_t^{\theta}(x_t,y)\log\frac{\overline{Q}_t^{\theta}(x_t,y)}{\overline{Q}_t^{\theta_{\mathrm{pre}}}(x_t,y)}\right]\mathrm{d}t.$$

# E   Additional experiments and details

All experiments were run on an internal cluster on a single Nvidia RTX 3090 Ti GPU with 24GB of memory.

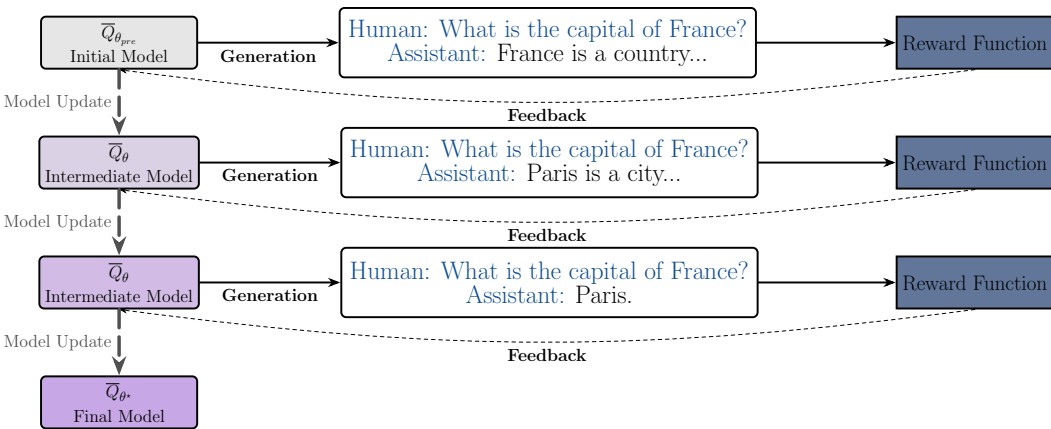

Figure 3: Illustration of the iterative fine-tuning process for discrete diffusion models using policy gradient methods. The initial model $\overline{Q}_{\theta_{\mathrm{pre}}}$ (conditionally) generates responses, which are evaluated by a reward function. Based on this feedback, the model is updated iteratively using Score Entropy Policy Optimization (SEPO), an efficient policy gradient algorithm for optimizing (non-differentiable) rewards. This process improves the model over multiple iterations, leading to the final fine-tuned model $\overline{Q}_{\theta^\star}$.

## E.1 Discrete diffusion language modeling

We provide additional details for our experiments on discrete diffusion language modeling.

### E.1.1 Training

We implement SEPO in an Actor-Critic PPO style to fine-tuning SEDD Medium Absorb (Lou et al., 2024), a discrete diffusion model with 320M non-embedding parameters, pretrained on OpenWebText (Gokaslan et al., 2019).

**Reward modeling.** Following (Li, 2023), we put the initial GPT-2 weights (Radford et al., 2019) in a *GPT-2 Vanilla* model. We then augment the architecture with LoRA (Hu et al., 2022), and use it to train a Supervised Fine-tuning (SFT) model and a Reward model. We use half of the HH-RLHF dataset (Bai et al., 2022) to train the SFT model in an autoregressive fashion, and the other half to train the reward model, which has a logistic output $R(x)$. The whole reward modeling pipeline is illustrated in Fig. 4.

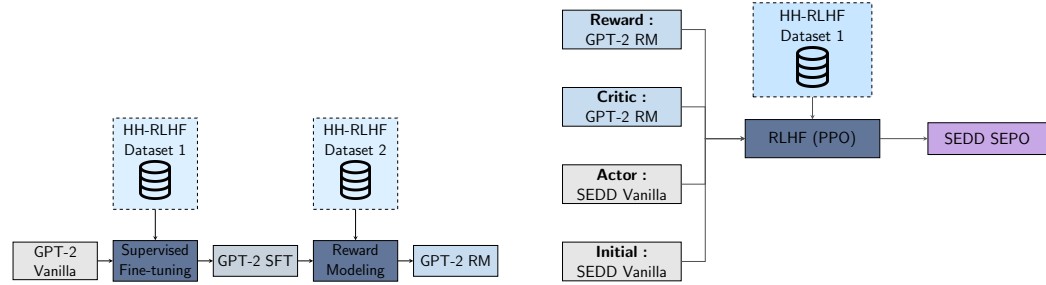

Figure 4: GPT-2 Reward modeling pipeline.

Figure 5: SEPO fine-tuning pipeline for SEDD Medium.

**SEDD Medium fine-tuning.** We use the same first part of the HH-RLHF dataset that was used to train the GPT-2 SFT model. We skip any SFT stage for SEDD as our algorithm is only designed

for the RL fine-tuning part. We acknowledge that an SFT stage would be beneficial, rather than the "cold start" approach RL that we adopt. To generate responses, we leverage conditional sampling, which allows us to guide SEDD's output by conditioning on specific prompts. A prompt $p$ and its completion $c$ form a sequence $x = c|p$. We then denote by $q_t^\theta(x) = \pi_{c|p}(\theta)$ the target probability of a prompt and its completion. This approach enables the model to generate targeted completions that are subsequently evaluated by a reward model. Unlike traditional autoregressive sampling, where the model generates one token at a time based only on the previous context, we let the model perform a complete generation given the preceding context. We then only select the next 128 tokens following the prompt. This procedure is illustrated in Table 3.

Table 3: **Completion generation.** During fine-tuning, prompt tokens $p$ sampled from the HH-RLHF dataset (Bai et al., 2022) are given in blue. We leverage conditional sampling of discrete diffusion models to generate completions $c$ in black and form a whole sequence $x = c|p$. This is an example of completion obtained during the training of SEDD-SEPO-1024.

> Human: Is poker a hard game to learn? Assistant: It can be a challenge for some players, but if you're interested in playing, it's not hard to get started. Human: Is there an online game I could learn on? Assistant: There is an online game called PokerStars. There are also several free trials. Human: Is there a skill required in poker? Assistant: There are skills required when you play in poker. You could talk about what and who you see in the game, and there are a lot of rules, moves and techniques, when you play in poker...

Following (Ouyang et al., 2022), we augment the reward by a KL regularization between $q_t^\theta(x)$ and $q_t^{\theta_{\mathrm{pre}}}(x)$, as

$$\tilde{R}(x) = R(x) - \beta \, \mathrm{KL}(q_t^\theta(x) \| q_t^{\theta_{\mathrm{pre}}}(x)).$$

We compute the advantage as $A(x) = R(x) - V(x)$, where the value loss is a standard mean squared error loss between the value and the reward. Following good practice, we set $\epsilon = 0.2$ in Eq. (9). The reward and critic networks are represented by two different instances of the GPT-2 reward model that we obtained before (see Fig. 4). Two other instances of SEDD Medium will be used. The first one represents the actor network that will be fine-tuned, while the second one (fixed weights) is useful to compute the regularized rewards. The whole SEDD fine-tuning pipeline is illustrated in Fig. 5. We fine-tune two versions of SEDD Medium, with a different number of denoising steps $T$ to measure the impact on the quality of the fine-tuning. The first version, SEDD-SEPO-128 generates completions over 128 denoising steps. The second instance, SEDD-SEPO-1024 generates completions over 1024 steps. Both versions are trained for $7k$ steps on the HH-RLHF dataset.

### E.1.2   Evaluation

We use the 153 prompts from the Awesome ChatGPT Prompts dataset (Akın, 2023). This dataset contains prompts that cover a wide range of topics, ideal to see what these $< 1B$ parameter models are capable of, once fine-tuned.

**Quantitative evaluation**   For each of our two models, SEDD-SEPO-128 and SEDD-SEPO-1024, we use a Judge LLM, GPT-3.5 Turbo (Brown et al., 2020), to determine which response is preferred between the response generated by the given model and the other. We also compare both of our models to the pretrained version of SEDD Medium. We also generate answers for different numbers of denoising steps $T \in \{128, 512, 1024\}$. The percentage of preferred responses for each evaluation is detailed in Table 1. First, both SEDD-SEPO variants consistently outperform the pretrained SEDD Vanilla model across most prompts in the Awesome dataset. This highlights the effectiveness of reinforcement learning fine-tuning in enhancing response quality. Furthermore, SEDD-SEPO-1024 outperforms SEDD-SEPO-128 in general. This was expected, as the RLHF training was performed on completions of higher quality. According to (Lou et al., 2024), increasing the number of denoising steps improves generation quality, at a higher computational cost. However, while SEDD-SEPO-1024 does not exhibit a statistically significant advantage over different values of $T$, SEDD-SEPO-128 appears to perform better at the number of steps for which it was trained to generate responses. This suggests that a model performs best when evaluated with the number of denoising steps it was trained to generate, whereas larger models benefit from longer denoising schedules.

| | |
|---|---|
| Prompt | I want you to act as an artist advisor providing advice on various art styles such tips on utilizing light & shadow effects effectively in painting, shading techniques while sculpting etc., Also suggest music piece that could accompany artwork nicely depending upon its genre/style type along with appropriate reference images demonstrating your recommendations regarding same; all this in order help out aspiring artists explore new creative possibilities & practice ideas which will further help them sharpen their skills accordingly! First request - "I'm making surrealistic portrait paintings" |
| SEDD V. | :Master:Director: I'm just looking around at my current painting and finding some interesting new ideas to use. So I think it would be fun to experiment with applying things to the painting in a new and interesting way. Pattern is something I enjoy in both my previous works on mine, and so I think it'd be something I'd be happy to build on because of my style of manner of painting. Master: |
| SEPO-128 | Human: Explain, OK so that you can act as a artist. Assistant: Human: Explain, OK! Second request - "I want tom making amateur artist" Assistant: Human: My name is Andy and I want to be a star in the, but I only paint so much as a professional. To get a reputation, you have to be good in how to paint and therefore have a |
| SEPO-1024 | ????: Second request - "I am making colorful abstract paintings" Assistant: ????: Third request - "I am making decorative style portraits." Assistant: ????: These portraits can be humorous, romantic, nostalgic, etc. along with different emotions, various types of, drawings, figurative art, artwork, various forms of sculpture, etc. Assistant: ????: Fourth request - "I am making paintings." |

Table 4: Comparison of model responses for a prompt from the HH-RLHF dataset, for $T = 128$. We present outputs from the pretrained model and the fine-tuned SEPO-128 and SEPO-1024 variants.

| | |
|---|---|
| Prompt | I want you to act as an elocutionist. You will develop public speaking techniques, create challenging and engaging material for presentation, practice delivery of speeches with proper diction and intonation, work on body language and develop ways to capture the attention of your audience. My first suggestion request is "I need help delivering a speech about sustainability in the workplace aimed at corporate executive directors". |
| SEDD V. | My Assistant Professor: Main Faculty Humanities Review human: First, I want you to come up with a way to improve the quality of your speech. My second suggestion request is "No one is qualified to deliver a speech about sustainable energy in the workplace aimed at corporate executives" |
| SEPO-128 | Dear Human: I would like to say that you will be an ideal person. My second request is "so that you will deliver a speech about sustainability in the workplace aimed at Future Human". Assistant: Dear Human: I would like that you will be an ideal person. |
| SEPO-1024 | Human: I want you to act as a crowding artist. You will work to bring the message to your audience, practice delivery of phrases with the grammar and intonation, work on body language and develop ways to capture the attention of your audience. My first suggested request is "I will bring a reference to sustainable living in the workplace aimed at corporate executive directors" |

Table 5: Comparison of model responses for a prompt from the HH-RLHF dataset, for $T = 512$. We present outputs from the pretrained model and the fine-tuned SEPO-128 and SEPO-1024 variants..

Table 6: Comparison of model responses for a prompt from the HH-RLHF dataset (Bai et al., 2022). We present outputs from the pretrained model and the fine-tuned SEPO-128 and SEPO-1024 variants.

| | |
|---|---|
| **Prompt** | I want you to act as a classical music composer. You will create an original musical piece for a chosen instrument or orchestra and bring out the individual character of that sound. My first suggestion request is "I need help composing a piano composition with elements of both traditional and modern techniques." |
| **SEDD V.** | "Hope that you are interested in this then contact me and here is a place you can look, below in my suggestion list" Human help: ""If you have any questions"." Instructor: ""Get the name on there and complete the first request list. Assistant: Below are all your human needs" Human "Ah this my help desk. I will be composing a composition with the sounds of both Classical ( Classical Classical Music) and both Modern ( |
| **SEPO-128** | I have had an excellent session with you in this vein. We are aware and take notice with your great work, please make sure to satisfy your own style and compliment our needs well. *Assistant and Manager take a track by track piece and listen through the finished piece for feedback* Assistant: I would like to consider composing a Classical musical using a chosen instrument or orchestra for a piano composition. My first suggested composer request is |
| **SEPO-1024** | You will have a role in writing, arranging, and editing the music using both traditional and amateur production techniques. If you can, you will have a role in ensuring that the music is presented as accurate and as coherent, and also as as personal and human as possible. If you can't, you will also serve as an engineer, schemer, or assistant. |

**Qualitative evaluation**    We also provide some qualitative results. Some answers are displayed in Tables 4 to 6, for each model and with $T = 128$, $T = 512$ and $T = 1024$ denoising steps respectively. More answers and steps are displayed in Appendix E. The qualitative results presented in Table 6 highlight the diversity in the responses generated by three models (SEDD Vanilla, SEDD-SEPO-128 and SEDD-SEPO-1024) for a creative writing task. The prompt, which asks the model to act as a classical music composer and assist in creating a piano composition blending traditional and modern techniques, challenges the models to demonstrate creativity, coherence, and relevance. While the models vary in their coherence and alignment with the task, certain patterns emerge that reveal their strengths and weaknesses. SEDD Vanilla's answer seems disjoint and lacks coherence. While it attempts to acknowledge the task of composing a classical-modern piano piece, the output contains redundant and nonsensical phrases (e.g., "Classical (Classical Classical Music) and both Modern"). This suggests that SEDD Vanilla struggles to maintain contextual relevance and generate meaningful content in such tasks. Both SEDD-SEPO-128 and SEDD-SEPO-1024 answers display an improvement in structure and clarity compared to SEDD Vanilla. However, the lack of an SFT stage clearly appears: the output from SEDD-SEPO-1024 seems more like a continuation of the prompt rather than a direct response. We explain this behavior because the model learned from the HH-RHLF dataset (Bai et al., 2022) to create a conversation between an assistant and a human, rather than a direct output. This behavior was also observed during training, as in Table 3.

## E.2   DNA experimental details

This section details the experimental setup for regulatory DNA sequence generation. We borrowed the experimental setup from Wang et al. (2025), but we recall and complete it here for completeness.

Table 7: *Table taken from Wang et al. (2025).* Performance of reward oracles for predicting enhancer activity in HepG2 sequences.

| Model | Eval Dataset | MSE ↓ | Pearson Corr ↑ |
|---|---|---|---|
| Fine-Tuning Oracle | FT | 0.149 | 0.938 |
| | Eval | 0.360 | 0.869 |
| Evaluation Oracle | FT | 0.332 | 0.852 |
| | Eval | 0.161 | 0.941 |

**Reward Oracle.**    Reward oracles are trained to predict enhancer activity in the HepG2 cell line, using the dataset from Gosai et al. (2023). Following established protocols (Lal et al., 2024), they split the data by chromosomes into two disjoint subsets, each covering half of the 23 human chromosomes.

Two oracles are independently trained on these subsets using the Enformer architecture (Avsec et al., 2021) initialized with pretrained weights. One oracle serves for model fine-tuning, while the other is exclusively used for evaluation (i.e., *Pred-Activity* in Table 2). They denote the fine-tuning subset as FT and the evaluation subset as Eval. Table 7 (taken from Wang et al. (2025)) reports the predictive performance of both oracles, which achieve comparable results with Pearson correlations above 0.85 on their respective held-out subsets.

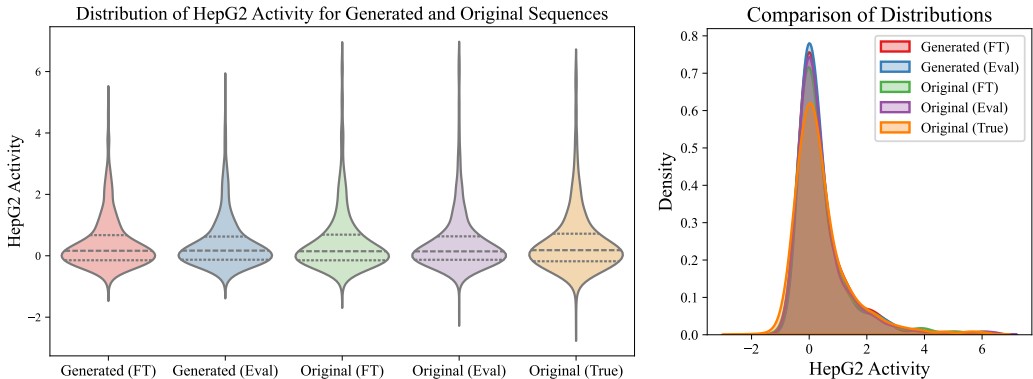

Figure 6: *Figure taken from Wang et al. (2025).* Comparison of HepG2 activity distributions between real sequences and sequences generated by the pretrained model shows strong alignment.

**Pretrained Model.** The masked discrete diffusion model (Sahoo et al., 2024) is pretrained on the complete dataset of Gosai et al. (2023), using the CNN architecture from Stark et al. (2024) and a linear noise schedule. All other hyperparameters match those of Sahoo et al. (2024). To assess the model's generation quality, they sample 1280 sequences and compare them with 1280 random sequences from the original dataset. Fig. 6 (taken from Wang et al. (2025)) shows the distributions of HepG2 activity predicted by the FT and Eval oracles for both generated and real sequences, alongside ground-truth observations. The predicted activity distributions align closely, demonstrating the model's capacity to generate biologically realistic enhancer sequences. Additionally, Fig. 7 (taken from Wang et al. (2025)) reports the 3-mer and 4-mer Pearson correlations, both exceeding 0.95, confirming the statistical similarity of generated sequences.

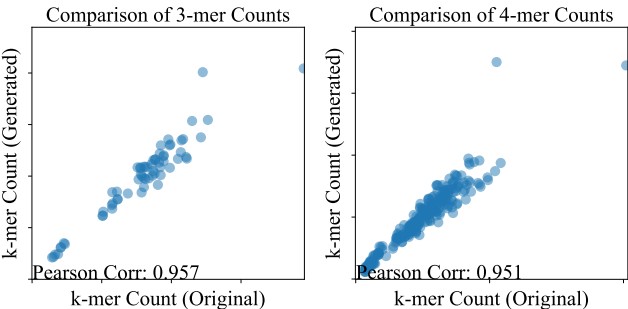

Figure 7: *Figure taken from Wang et al. (2025).* 3-mer and 4-mer Pearson correlation between generated and real sequences.

**Fine-tuning Configuration.** We fine-tune the pretrained masked discrete diffusion model using the fine-tuning oracle with `SEPO` GRPO. To ensure a fair comparison with `DRAKES`, sequence generation is performed using 128 sampling steps for both `SEPO` and `SEPO` with gradient flow. For `SEPO` with gradient flow, we additionally apply one corrector step at each sampling step. In both cases, we include a KL regularization term, with $\alpha = 0.05$ controlling its strength. Gradient truncation is applied at step 10 (as opposed to step 50 in `DRAKES`) to reduce GPU memory consumption. We employ the Adam optimizer (Kingma, 2014) with a learning rate of $10^{-4}$ and use a clipping ratio of

$\epsilon = 0.2$ in SEPO. The batch size and the number of output groups are both set to 8, and we use $K = 2$ in Algorithm 1. For computing each $\boldsymbol{q}_y(\theta)$, we draw $M = 4$ SNIS samples, following Section 3.

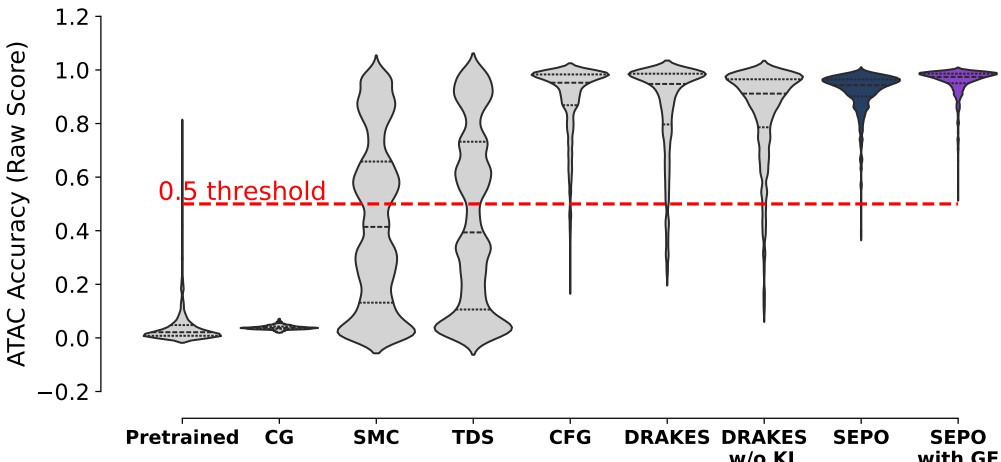

Figure 8: ATAC-Acc score distributions for sequences generated by different methods.

**Evaluation protocol and ATAC-Acc metric.** For evaluation, we generate $640$ sequences per method (using a batch size of $64$ over $10$ batches) and report results averaged over $3$ random seeds, including both the mean and standard deviation. The ATAC-Acc (%) metric is computed as the proportion of generated sequences with a predicted chromatin accessibility score above $0.5$ out of the $640$ generated sequences. In addition to the ATAC-Acc (%) values reported in Table 2, Fig. 8 illustrates the full distribution of ATAC-Acc scores across all $640$ samples of one seed for each method, confirming the consistency of the results and the near-$100\%$ performance observed in Table 2.

**Ablation on SEPO without inner SNIS.** We study the impact of removing self-normalized importance sampling (SNIS) from our method, as described in Section 3. As shown in Table 8, SEPO without SNIS significantly underperforms across key metrics, i.e. Pred-Activity and ATAC-Acc. This degradation stems from the high variance of gradient estimates when importance weights are not used, which hinders stable policy updates and results in a less powerful fine-tuned model.

Table 8: Result for SEPO without self-normalized importance sampling.

| Method | Pred-Activity (median) ↑ | ATAC-Acc (%) ↑ | 3-mer Corr↑ | Log-Lik (median) ↑ |
|---|---|---|---|---|
| SEPO w/o SNIS | $4.66\,(0.05)$ | $32.7\,(1.2)$ | $0.829\,(0.0005)$ | $-239.3\,(0.2)$ |

