# OpenReview forum: "Fine-Tuning Discrete Diffusion Models with Policy Gradient Methods"
_NeurIPS.cc/2025/Conference — NeurIPS 2025 poster_

### Official Review · Reviewer_sGKa · 2025-07-02

**Clarity:** 2
**Significance:** 2
**Originality:** 2
**Rating:** 3
**Confidence:** 4

**Summary:**

This paper proposes "bandit" policy gradient algorithms for discrete diffusion models in the concrete score framework. The proposed method SEPO is tested  on DNA fine-tuning and natural language tasks and compared against other benchmarks. It also introduces a gradient flow alternative that improves sample quality at a higher complexity. Convergence bounds and derivations of the method are provided.

**Questions:**

1. Regarding weakness 2, what is the justification for only summing along the neighbors?
2. Why use a trajectory of length one instead of the more common MDP approach for gradient policies?
3. How do the proposed methods compare with the existing ones in terms of compute?

**Ethical Concerns:**

["NO or VERY MINOR ethics concerns only"]

**Final Justification:**

I am increasing my score to 3, as the authors have acknowledged the issue of equating $s_\theta$ this the ratios of probabilities that $s_\theta$ induces, and have proposed to instead state that this is an approximation, and use the notation $f_\theta$ for the real quantity. I cannot give the paper a higher rating, as their proposed modifications, requires large changes for the camera ready version and the assumptions become strong: that $s$ and $f$ are close and that we can set $\nabla_\theta \log f$ to zero for non neighbor ratios.

**Limitations:**

The computational complexity of the the inner importance sampling steps is acknowledged in the conclusion. A separate limitation section is not provided.

**Paper Formatting Concerns:**

I did not notice any major formatting issues.

**Quality:**

2

**Strengths And Weaknesses:**

**Strengths**

1. The problem tackled by this paper is interesting and important.
2. The ideas and proof strategies are interesting.
3. The experiments are interesting and compare with multiple baselines.

**Main weaknesses**
The main weaknesses of this paper concern the validity of the proofs/statements and the clarity of the paper:
1. Many of the proofs/statements (Theorem 3.1, Theorem 3.2, SNIS line 145) hinge on the claim that ${s_y(x,\theta,t)}$ =$ \frac{q_\theta(y)}{q_{\theta}(x)}$. This is not necessarily the case. If $s(x,\theta,t)_y$ has modeled the ratios $p(y)/p(x)$ perfectly, then it is correct, otherwise it is not necessarily true. This is a very important fact, and therefore either this should be stated as an assumption or be proven. Proving it would imply that the bound given in Lou et al 2024, becomes an equality, so it is unlikely to be true.
2. $q_t^\theta(y)$ in Theorem 3.1 comes from $Z_\theta$, which is the sum over all states $y$. Therefore, as properly written in the paper, the inner sum is taken over all states, not just neighboring ones for which we have the learned scores. When sampling in SEDD this is okay, because the transition-rate matrix is sparse, and therefore due to multiplication, to go backwards, we only need to model neighbor scores. However, in the formula of Theorem 3.1 and 3.2, the transition-rate matrix is not there to simplify the equation. I would have checked the implementation but the code is not provided.
3. The SNIS approach proposed appears to be computationally expensive, and in addition, regarding line 145, the original problem was that we do not have the values $q_t^\theta (y)$. It is unclear why this approach is sufficient.
4. The paper is a bit unclear in the sense based on the way it is written, it leads you to expect a usual gradient policy method, utilizing trajectories with multiple steps, but only the end of the trajectory is utilized in the loss. In [1] RWR is introduced, but it also proposes multistep DDPO, similarly to [2] (in the continuous case). Reading the introduction, and abstract one is likely to expect something similar.
5. The difference in compute and time is not provided between the proposed method and other methods. One would expect that the inner loop is expensive to calculate.

**Minor weaknesses**
1. Line 81 and 87, should be $Q_t(y,x)$ instead of $Q_t(x,y)$.
2. The cummulative reward from time $t$ to 0 in [1], is the constant $r(x_0,c)$ and not 0.
3. The lower left results in Table 1 are not necessary.

[1] TRAINING DIFFUSION MODELS WITH REINFORCEMENT LEARNING, Black et al, 2024

[2] Diffusion Model Alignment Using Direct Preference Optimization, Wallace et al, 2023

---

> ### Author Rebuttal · Authors · 2025-07-29
>
> ## Main answer
>
> ### Main weakness
>
> We thank Reviewer sGKa for the detailed review and for highlighting the importance of the problem and the contributions of our paper, including the theoretical formulation, gradient flow variant, and experimental validation.
>
> We appreciate the reviewer’s critical feedback regarding the clarity and technical interpretation of our derivations and estimator design. Several concerns appear to stem from misunderstandings of the setting or modeling assumptions, which we address point by point below. We are confident that clarifying these points will significantly improve the paper and its accessibility.
>
> > Many of the proofs/statements (Theorem 3.1, Theorem 3.2, SNIS line 145) hinge on the claim that s_y(x,\theta,t) = \frac{q_\theta(y)}{q_{\theta}(x)} This is not necessarily the case. If s_y(x,\theta,t) has modeled the ratios $\frac{p(y)}{p(x)}$ perfectly, then it is correct, otherwise it is not necessarily true. This is a very important fact, and therefore either this should be stated as an assumption or be proven. Proving it would imply that the bound given in Lou et al 2024, becomes an equality, so it is unlikely to be true.
>
> Thank you for raising this important point. We respectfully disagree and believe there has been a misunderstanding.
> The identity $s_\theta(x)= \frac{q_\theta(y)}{q_\theta(x)}$ is not an assumption, but follows from the **completeness of the concrete score**, proved in Theorem 1 of [1] (a previous publication by co-authors of Lou et al. 2024 [2] where the notion of concrete score is introduced).
>
> Specifically, as established in [1], the concrete score _for a distribution_ $p$ is defined as
> $$c_p(x)y = \frac{p(y)}{p(x)} - 1,$$
> so that the quantity ${s_\theta(x)}y = {c_{q_\theta}(x)}y + 1$ represents $\frac{q_\theta(y)}{q_\theta(x)}$.
>
> The result of Theorem 1 in [1] is the completeness of the concrete score representation: if two concrete scores match, then the underlying distributions match (i.e., $c_{p_{\theta}} = c_{p_{\text{data}}} \Rightarrow p_\theta = p_{\text{data}}$). This guarantees that the parametrized score representation $s_\theta(x)$ and the underlying distribution $q_\theta(x)$ have equivalent expressive power, and makes the "claim" perfectly valid.
>
> Note that this result is **distinct from the notion of consistency**, which is addressed separately (in Theorem 2 of [1] for the CSM loss and Proposition 3.2 of [2] for the score entropy loss), and refers to whether a loss function recovers the correct score function in the infinite data/computation limit. Importantly, our statements and estimators **do not assume** that the model has already learned the true score, but only rely on this well-established structural equivalence between score ratios and the underlying distribution.
>
> Regarding the reviewer’s final comment, we agree that the ELBO bound in Lou et al. (Theorem 3.6) only becomes an equality when the learned score matches the ground truth: a standard condition related to **consistency**, not completeness. Our "claim" does not require or imply this equality; rather, we use the mathematical identity known as completeness to define and interpret our estimators.
>
> Following the reviewer's comment, we will add a remark in the revised paper to clarify this distinction between completeness, consistency, and estimator design to help the readers.
>
> > $q_t^\theta(y)$ in Theorem 3.1 comes from $Z_\theta$, which is the sum over all states $y$. Therefore, as properly written in the paper, the inner sum is taken over all states, not just neighboring ones for which we have the learned scores. When sampling in SEDD this is okay, because the transition-rate matrix is sparse, and therefore due to multiplication, to go backwards, we only need to model neighbor scores. However, in the formula of Theorem 3.1 and 3.2, the transition-rate matrix is not there to simplify the equation. I would have checked the implementation but the code is not provided.
>
> We appreciate the reviewer pointing this out. Indeed, the theoretical formulation of the gradient involves a sum over the full state space. However, as in [2] (that also provide a theoretical formulation of the loss on the whole state space), we exploit the local construction of the score network $s_\theta$ that appears in Theorem 3.1 and 3.2, where only all ratios between sequences with Hamming distance 1 are modeled. This allows us to compute the required score functions locally and restrict the summation in practice to the neighborhood, exactly as they do in [2]. We will include a link to the public code repository at the camera ready version.
>
> > The SNIS approach proposed appears to be computationally expensive, and in addition, regarding line 145, the original problem was that we do not have the values $q_t^\theta(y)$. It is unclear why this approach is sufficient.
>
> We agree that SNIS introduces additional computational cost during training. As also raised by Reviewer 3dgX, we will include in the revised version a dedicated discussion outlining the computational trade-offs, practical scaling behavior, wall-clock times (table below) and the impact on training efficiency.
>
> As explained in line 145, we do not have the value of the proposal distribution $q_t^\theta(z_i)$, but we do have the value of $q_t^\theta(z_i)/q_t^\theta(y)$ through $s_\theta(z_i,T-t)_y$. In other words, we can evaluate the proposal $q_t^\theta$ on each $z_i$ up to a normalization constant. SNIS is a statistical method that has been developed when the proposal of Importance Sampling (IS) can only be evaluated up to a constant : normalization cancels the constant. If we knew perfectly $q_t^\theta(z_i)$, SNIS would be useless because IS would be sufficient.
>
> > The paper is a bit unclear in the sense based on the way it is written, it leads you to expect a usual gradient policy method, utilizing trajectories with multiple steps, but only the end of the trajectory is utilized in the loss. In [1] RWR is introduced, but it also proposes multistep DDPO, similarly to [2] (in the continuous case). Reading the introduction, and abstract one is likely to expect something similar.
>
> Both references suggested by the reviewer fall within the scope of our general framework, and their formulations are covered by our design of the reward $R_t$ and policy gradient objective. In fact, one can reformulate our denoising process $q^\theta_t$ as a $T_0-$horizon MDP:
>
> \begin{align*}
>     \text{For all } t \in \{0,\dots, T_0\},\quad \mathcal{S} &= \mathcal{X}, \quad \mathcal{A} = \mathcal{X}, \quad \text{Policy } q^\theta_t,\\
>     S_t &= x_{T_0-t}, \quad A_t = x_{T_0-1-t}, \quad R_{t}=
>     \begin{cases}
>         R(x_0) & \text{if } t = T_0, \\
>         0 & \text{if } t < T_0.
>     \end{cases}
> \end{align*}
> with known transition dynamics: $P(S_0) = p_{ref}, \quad {P(S_{t+1} \mid S_t, A_t) = \delta({S_{t+1} = A_t})}.$
>
> Nonetheless, we acknowledge that this connection may not have been immediately evident. To enhance clarity, we will incorporate a multi-step MDP reformulation, following the structure used in the first reference, to facilitate comparison with existing RLHF-style diffusion methods.
>
>
> > The difference in compute and time is not provided between the proposed method and other methods. One would expect that the inner loop is expensive to calculate.
>
> It is true that SNIS involves additional computation due to the need to compute transition probabilities. Specifically, for each of the $N$ initial samples $(x_1, \ldots, x_N)$, we perform a batched forward pass through the score network to evaluate the transition probabilities required for SNIS. This leads to a moderate increase in compute.
>
> To quantify this, we now include wall-clock training time comparisons between SEPO and DRAKES in the following table (included in the revised version). These measurements were taken using the same hardware (a single GPU NVIDIA GeForce RTX 3090 with 24GB VRAM) and batch sizes to ensure fair comparison.
>
> | Method                | KL Trunc. Step | # Epochs | Batch Size/GRPO batch size | Runtime (hh:mm) | Time/Epoch (hh:mm:ss) | Final Pred-Activity |
> |------------------------|-------------|-----------|-------------|---------|-------------|----------------|
> | SEPO                   | 10          | 14       | 8 / 8       | 00:19   | 00:01:21    | 7.55         |
> | SEPO (gradient flow)   | 10          | 8       | 8 / 8       | 00:24   | 00:02:59    | 7.64         |
> | SEPO (no SNIS)         | 10          | 199       | 8 / 8        | 00:10   | 00:00:03    | 4.66          |
> | Drakes (no KL)         | -           | 127       | 8 / -       | 00:26   | 00:00:12    | 6.41          |
> | Drakes (with KL)       | 50          | 66       | 8 / -      | 00:31   | 00:00:28    | 5.70         |
>
> Even with this limited compute, it was possible to finetune models in a reasonable time.
>
> ### Minor weakness
>
> Minor weakness 1 : We thank the reviewer for noticing that typo. It will be addressed in the revision.
>
> Minor weakness 2 : Our denoising process $q_t^\theta$ is defined as an approximation of $p_{T-t}$ (not $p_t$). This is simply how we denotes our denoising process and, consequently, how we define the reward function $R_t$ whose index $t$ correpspond to that of $q_t^\theta$. We believe that, in light of our clarification regarding main weakness 4, this point will now be clearer to the reviewer.
>
> Minor weakness 3 : We prefer to leave the result for clarity as it does not take more space, but we are open to removing it if the reviewer believes it is necessary.
>
>
> ## Answer to questions
>
> The three questions were adressed above.
>
> ---
>
> We hope that our clarifications and revisions have addressed the concerns raised, and would be grateful if the reviewer considers updating its score.
>
> [1] Concrete Score Matching: Generalized Score Matching for Discrete Data, Meng et al. (2023)
>
> [2] Discrete Diffusion Modeling by Estimating the Ratios of the Data Distribution, Lou et al. (2024)

---

> > ### Comment · Reviewer_sGKa · 2025-08-03
> >
> > I thank the authors for the additional work and their response.
> >
> > Unfortunately, my main concerns related to the technical correctness of the paper were not in my opinion properly addressed. I summarize everything below:
> >
> > 1. Regarding the assumption $s(x,\theta,t)_y=\frac{q_t^\theta(y)}{q_t^\theta(x)}$: From [1] it is not true that $s(x,\theta,t)_y$ equals $\frac{q_t^\theta(y)}{q_t^\theta(x)}$. All Theorem 1 states is that (under certain conditions), if the concrete scores defined as
> >  $c_q^\theta = \frac{q_t^\theta(y)}{q_t^\theta(x)}-1$ are such that they equal the true concrete scores, then the probability flow they define equals the true probability flow. On the other hand, $s(x,\theta,t)_y$ is the neural approximation of the true score. The quantity $c_q^\theta -1$ is not the same as modeled scores. As I mentioned in the review, however, they do agree when the model learns the scores perfectly, in which case: $s(x,\theta,t)_y=\frac{q_t^\theta(y)}{q_t^\theta(x)}=c_q^\theta -1=\frac{q_t(y)}{q_t(x)}=c_q -1$. But assuming the model has perfectly learned the scores is a **very** strong assumption. Theorem 1 (and its proof) does not say anything about the case in which $c_q^\theta\neq c_q$. Similarly Theorem 2 therein simply states that in the limit of infinite data and infinite model capacity, the optimal parameters recover the true scores. It does not make any statement about the relation of the imperfectly learned scores and the ratios of probabilities that such an imperfectly learned score induces.
> >
> >  In the continuous diffusion case, in Theorem 2 of [2], it is stated that the KL bound becomes and equality if the learned score models the true score. Table 2 in that paper, shows that in reality the bound and the true NLL differ. In the case of discrete diffusion, in Theorem 2 of [3] it is shown that the KL bound, and therefore the bound in [4] becomes an equality if $s(x,\theta,t)_y=\frac{q_t^\theta(y)}{q_t^\theta(x)}$. They show that one such case is when the ratios have been modeled perfectly: $s(x,\theta,t)_y=\frac{q_t(y)}{q_t(x)}$. The authors can calculate the bound on cross entropy (Theorem 3 of [3]) and the true cross entropy on a small experiment say, sequence of length 2 in a vocabulary of length 2, and evaluate the difference. They will most likely find that the bound is not tight, implying that $s(x,\theta,t)_y \neq \frac{q_t^\theta(y)}{q_t^\theta(x)}$.
> >
> > All things considered, the burden of proof lies with the authors. I remain open-minded and will revise my evaluation if they provide a fully rigorous proof that establishes:
> > $s(x, \theta, t)_y = \frac{q_t^\theta(y)}{q_t^\theta(x)}$,
> > where $s(x, \theta, t)_y$ denotes the scores modeled by the neural network, and $\frac{q_t^\theta(y)}{q_t^\theta(x)}$ represents the ratio of probability flows induced by these learned scores.
> >
> > 2.  Regarding Theorems 3.1 and 3.2: as I mentioned in my review, in the context of Equation (1) in your paper, the matrix $Q$ is sparse, which justifies the need to consider only the ratios between neighboring states. However, in Theorems 3.1 and 3.2, the inner sum is taken over all $y \neq x$, where $y \in \mathcal{X}$, and $\mathcal{X}$ is defined (in line 70) as the entire grid (or space). Neither theorem provides a justification for restricting attention to neighboring terms in $\nabla_\theta \log s(x, \theta, T - t)_y$, especially given that there is no $Q$ matrix present to exploit sparsity and simplify the expression accordingly.
> >
> > These two points are my primary concerns, as they are central to the arguments made throughout the paper.
> >
> > The following two points are of a more minor nature:
> >
> > 3. While it is clear that your formulation is a special case, my comment was primarily about the motivation behind this specific choice. Specifically, why set $R_t = 0$ for all $t \neq T_0$? Given that this assumption is a focal point of the paper, a more thorough justification would be appreciated.
> >
> > 4. Thank you for providing the additional experiments. Although they do not cover all models presented in the paper, they offer useful information regarding compute requirements. As expected, the method incurs a computational slowdown when using SNIS; however, one could argue that this trade-off is justifiable.
> >
> > I thank the authors for their time and remain open to further discussions regarding points 1, 2, and 3.
> >
> > [1] Meng et al. Concrete Score Matching: Generalized Score Matching for Discrete Data
> > [2] Song et al. Maximum Likelihood Training of Score-Based Diffusion Models
> > [3] Haxholli et al. Efficient Perplexity Bound and Ratio Matching in Discrete Diffusion Language Models
> > [4] Lou et al. Discrete Diffusion Modeling by Estimating the Ratios of the Data Distribution

---

> > > ### Author Response · Authors · 2025-08-04
> > > **Answer (1/2) by authors**
> > >
> > > **(Answer to points 1 and 2 (1/2) )**
> > >
> > > We sincerely thank the reviewer for the detailed and insightful feedback.
> > >
> > > 1.
> > > Now that the reviewer has clearly re-explained the issue, we fully understand what was confusing in our initial proofs. We also sincerely apologize for not fully understanding the reviewer's concerns initially. The reviewer is correct in this assessment, and we greatly appreciate the thorough and patient clarification provided.
> > >
> > > To explicitly address the ambiguity, we introduce the following notation:
> > >
> > > $$f(\theta,x,y,t) := \frac{q_t^\theta(y)}{q_t^\theta(x)}$$
> > >
> > > which precisely represents the ratio of probabilities induced by the probability flow. Throughout our proofs, we will carefully distinguish this theoretical quantity from its neural network approximation $s_\theta(x,t)y$, and will replace every incorrect occurrence of $s_\theta(x,t)_y$ with $f(\theta,x,y,t)$.
> > >
> > > The theoretical, exact gradient (e.g. for Theorem 3.1) in its most general form is then written as:
> > >
> > > $$\nabla_\theta\ell_t(\theta) =  \sum_{x\in\mathcal{X}}q^\theta_t(x)R_t(x)\sum_{\substack{y \in \mathcal{X} \\ y \neq x}}{q^\theta_t(y)\nabla_\theta\log f(\theta,x,y,T-t)}.$$
> > >
> > > since all mathematical derivations remain valid for $f$ under the substitution of $s_\theta(x,t)_y$ with $f(\theta,x,y,t)$ in the proofs.
> > >
> > > We will clearly articulate in the revised manuscript that this exact theoretical gradient is not practically accessible due to two main challenges: (i) the "probability flow" ratio $f(\theta, x, y, t)$ is unknown, and (ii) the summation extends over all states $y\neq x$, which is computationally infeasible.
> > >
> > > Therefore, in practice, we will explain that we replace the unknown true ratio $f(\theta, x, y, t)$ with its convenient neural approximation $s_\theta(x,t)_y$, leading to the following approximation of the gradient:
> > >
> > > $$\nabla_\theta\hat{\ell_t}(\theta) =  \sum_{x\in\mathcal{X}}q^\theta_t(x)R_t(x)\sum_{\substack{y: \\ s_\theta(x,T-t)y \text{ is defined}}}{q^\theta_t(y)\nabla_\theta\log s_\theta(x,T-t)_y}$$
> > >
> > > Here, note that the inner sum is defined on the only states $y$ for which the neural approximation $s_\theta(x,t)_y$ of $f(\theta, x, y, t)$ is explicitly available (we elaborate on this further in point 2.).
> > >
> > > We will emphasize these clarifications explicitly in the revised manuscript to ensure full rigor and transparency, and to directly address the reviewer’s valuable feedback.
> > >
> > > 2.
> > >
> > > As we explained above, In both Theorems 3.1 and 3.2, the exact gradient $\nabla_\theta\ell_t$ is written as a sum over every state $y\neq x\in \mathcal{X}$. However, in our implementation the neural network only ever produces a score estimate $s_\theta(x,t)y$ for those $y$ that differ from $x$ by a single token (i.e. the Hamming-distance-1 neighbors). No network outputs exist for any other pairs $(x,y)$, so as soon as we replace the true (but unknown) ratio $f(\theta,x,y,t)$ by its neural approxiamtion $s_\theta(x,t)_y$, all non-neighbor terms simply vanish as they are undefined.
> > >
> > > Of course, $\nabla_\theta\hat{\ell_t}$ is therefore an approximation of the theoretical true gradient $\nabla_\theta\ell_t$, and we will state this clearly as the reviewer has underlined.
> > >
> > > We will incorporate the resolutions to points 1 and 2 throughout the revised manuscript, and we once again thank the reviewer for their patience and insightful feedback, which have significantly strengthened the mathematical rigor of the paper. We hope these revisions fully address the reviewer’s concerns.

---

> > > > ### Author Response · Authors · 2025-08-04
> > > > **Answer (2/2) by authors**
> > > >
> > > > **(Answer to point 3 (2/2))**
> > > >
> > > > 3.
> > > >
> > > > We thank the reviewer for raising the question about our choice to set $R_t = 0$ at all intermediate steps $t<T_0$. In fact, this “terminal-only” reward design is standard in the RL-for-diffusion literature. The core reason (see the review [1] for a more detailed discussion) is that, until the final diffusion step, samples remain heavily corrupted by noise, making it difficult (and uncommon) to assign a per-step reward that reliably correlates with ultimate sample quality. Consequently, most existing methods compute the reward only at $t = T_0 = T$, when the sample is fully denoised. Accordingly, we adopt the same setup in our experiments. Other exisiting methods (e.g. [2]) perform denoising for only $T_0 \leq T$ steps, which directly motivated our formulation. We acknowledge that this is not clearly explained in lines 111–114, and will elaborate further in the revised manuscript, including the multi-step MDP formulation discussed previously with the reviewer.
> > > >
> > > > However, it is important to emphasize that our theoretical results (Theorems 3.1, 3.2, etc.) are derived for a completely general reward $\{R_t\}_{t\in [0,T]}$, not just the special case $R_t = 0$ for $t < T_0$. This generality means that one can plug in nonzero intermediate rewards (should future work devise meaningful per-step evaluation functions) without invalidating any of our proofs. We also note the quite recent work [3], which begins to address this problem.
> > > >
> > > > [1] Understanding Reinforcement Learning-Based Fine-Tuning of Diffusion Models: A Tutorial and Review, Uehara et al. (2024)
> > > >
> > > > [2] Diffusion-based Curriculum Reinforcement Learning, Sayar et al. (2024)
> > > >
> > > > [3] Towards Better Alignment: Training Diffusion Models with Reinforcement Learning Against Sparse Rewards, Hu et al. (2025)
> > > >
> > > > We thank the reviewer once again for their thorough and constructive review, which has greatly strengthened our manuscript.

---

> > > > > ### Comment · Reviewer_sGKa · 2025-08-04
> > > > >
> > > > > I thank the authors for the response.
> > > > >
> > > > > Regarding the more minor point 3, I do not think there is an RL 'standard' for diffusion. Different methods exist and continue to emerge. One of the main papers upon which this work builds (Black et al), expands RWR to DDPO, after introducing RWR. The following quotes are taken from their paper: "RWR relies on an approximate log-likelihood because it ignores the sequential nature of the denoising process, only using the final samples $x_0$" and "We present a
> > > > > general framework with an arbitrary reward function, motivated by our desire to optimize arbitrary
> > > > > downstream objectives". Other important methods exist which apply Direct Policy Optimization [1], and use multiple snapshots. Regardless, point 3 was a statement about clarity rather than a criticism of methodology.
> > > > >
> > > > > [1] Wallace et al. Diffusion Model Alignment Using Direct Preference Optimization.
> > > > >
> > > > > Regarding the important points 1 and 2, I thank the authors for acknowledging the issue. The promised changes would solve the technical incorrectness of the paper (although this would require quite a significant revision), but at the price of making the methodology not very well supported theoretically:
> > > > > 1. How far are $s$ and $f$ apart in practice? Is there a bound that would justify the substitution?
> > > > > 2. As the authors pointed out, the non-neighbor scores exist, they are not zero (and their derivative is likely to be non-zero), even though we do not know them. Without the matrix $Q$ to eliminate them, we simply have to set $\nabla_\theta \log s$ to zero for those ratios, and hope this is a good approximation, which likely is not ideal as the $\nabla_\theta \log  f$ is unlikely to be 0.
> > > > >
> > > > > Taking these points into account, I am raising my score to a 3, as the proposed revisions should address the technical flaws. However, I still cannot recommend acceptance, as the paper relies on assumptions that remain quite strong, and the necessary changes would significantly alter large portions of the work.
> > > > >
> > > > > I thank again the authors for the discussion, and I am glad our exchange could help strengthen the mathematical rigor of the paper.

---

> ### Author Response · Authors · 2025-08-05
>
> We thank the reviewer for engaging further in this discussion and for raising these important points. We are glad that our earlier response resolved the reviewer main technical queries, and we appreciate the opportunity to clarify a few additional matters.
>
>
> 1.
>
> Rather than deriving a hard bound, it is common in diffusion‐model theory to _assume_ that the error of the neural‐network score estimator is of order $\mathcal{O}(\epsilon)$, where $\epsilon$ is a small threshold.
>
> This assumption reflects both the expressive power of modern networks and the fact that they are trained to high accuracy, and it appears in several prior works in dicrete diffusion (cf Assumption 4.6 and 4.6' in [1] and Assumption 1 in [2].)
>
> 2.
>
> As written in line 151, the true gradient takes the form $\nabla_\theta\ell_t(\theta) = \mathbb{E}[x\sim q^\theta_t][R_t(x)g(x,\theta)]$. In practice, as discussed previously, we are computing $\nabla_\theta\hat{\ell_t}(\theta) = \mathbb{E}[x\sim q^\theta_t][R_t(x)\hat{g}(x,\theta)]$ where $\hat{g}(x,\theta) = \sum_{\substack{y: \\ s_\theta(x,T-t)y \text{ is defined}}}{q^\theta_t(y)\nabla_\theta\log s_\theta(x,T-t)_y}$.
>
> As noted by the reviewer, $\nabla_\theta \log f$ is unlikely to be zero for non-neighbours ratios : dropping those terms does introduce bias in the estimation of $g$ by $\hat{g}$. We will explicitly note this when defining $\nabla_\theta\hat{\ell_t}(\theta)$, and discuss how this bias arises.
>
> However, note that even with this additional bias, we achieve SOTA results against various methods in discrete diffusion. SEPO introduces a policy optimization objective that only depends on concrete score estimation, circumventing for example likelihood approximation that are done in [3,4]. Figuring out how to mitigate this additional bias in practice and potentially quantifying it theoretically are interesting directions for future works, and we will add this discussion in the conclusion section.
>
> Our intuition is that this bias is manageable, because the concrete score $c_p(x)$ (as its continuous space equivalent $\nabla\log p(x)$) is a local descriptor of how the probability mass is changing around $x$ : the summed contribution of $f(\theta,x,y,t)$ terms when $y$ is far apart from $x$ in the graph may have a small influence in how $\hat{g}$ approximates $g$. The strong empirical results seem to support this intuition. In fact, if this bias were large enough to spoil the gradient, our fine-tuning would fail; instead, we consistently observe strong gains and stable outputs once the model is finetuned.
>
> Additionally, we would also like to emphasize that while the time/epoch of our method is higher, it requires an order of magnitude less epoch to reach a higher score. So the total runtime is actually lower than SOTA method while reaching a much higher score. This is something we had not recorded prior to submission so we are grateful to the reviewer for suggesting this, which makes our experimental section and comparison with other methods even stronger.
>
> We thank once again the reviewer for their time, the constructive review and for prompting these clarifications. The feedback has substantially improved both the theoretical presentation and the empirical comparison in our manuscript.
>
> [1] How Discrete and Continuous Diffusion Meet: Comprehensive Analysis of Discrete Diffusion Models via a Stochastic Integral Framework, Ren et al (2024)
> [2] Convergence Analysis of Discrete Diffusion Model: Exact Implementation through Uniformization, Chen et al. (2024)
> [3] d1: Scaling Reasoning in Diffusion Large Language Models via Reinforcement Learning, Zhao et al. (2025)
> [4] wd1: Weighted Policy Optimization for Reasoning in Diffusion Language Models, Tang et al. (2025)

---

### Official Review · Reviewer_snkJ · 2025-07-02

**Clarity:** 3
**Significance:** 2
**Originality:** 2
**Rating:** 4
**Confidence:** 3

**Summary:**

This paper introduces SEPO, a novel policy gradient-based algorithm for fine-tuning discrete diffusion models using non-differentiable reward functions. The authors propose a theoretically grounded framework that adapts policy gradient techniques like PPO and TRPO to discrete diffusion settings, leveraging score entropy and importance sampling. Essential validations are conducted on DNA sequence design and language modeling tasks.

**Questions:**

1. Why are standard RLHF baselines such as PPO, DPO, or TRPO not included in the language modeling experiments, even though SEPO is proposed as a general-purpose policy gradient method?
2. How critical is the gradient flow sampling mechanism to SEPO’s performance? Would simpler sampling strategies yield comparable results?
3. Can the authors explain what the major difficulty of this algorithm (theoretically and experimentally) is beyond adapting existing policy optimization algorithms to a discrete-time formulation>

**Ethical Concerns:**

["NO or VERY MINOR ethics concerns only"]

**Final Justification:**

Overall the algorithmic contribution and empirical observations are good, which makes this work above the borderline. The technical flaw is vital though, and I am not sure if the authors can make a nice fix in the publication version if this work is accepted. From my understanding of general publication guidelines, it seems an undergoing-construction work should not be accepted unless the fix is done and is reviewed again. Therefore, I think this will be a complicated situation for a final decision.

**Quality:**

3

**Strengths And Weaknesses:**

Strength:

1. The paper presents a comprehensive theoretical formulation of SEPO, which adapts classic RL algorithms to a score-entropy-based discrete diffusion setting. The derivations, particularly Theorems 3.1 and 3.2, and the convergence result (Theorem 3.5), demonstrate solid mathematical rigor.
2. I think the inclusion of a gradient flow-based sampling strategy is a notable extension.
3. Experimental results on DNA sequence modeling demonstrate that SEPO (especially with gradient flow) outperforms prior methods in key metrics, such as enhancer activity and chromatin accessibility.

Weakness:

1. The baselines included in this work remain narrow. Critical missing comparisons include inference-time-only approaches like discrete SMC methods or newer masked diffusion fine-tuning techniques. For language modeling, standard RLHF baselines, such as DPO or PPO-tuned transformers, are almost a must-include for a fairer evaluation.
2. SEPO’s inner sampling and score estimation steps (especially SNIS) introduce notable computational overhead; relevant discussions are not included.
3. Technically there is a concern with the technical hardness of this work. Compared to existing works, the major work is done by replacing continuous diffusion models with discrete diffusion models and the optimization algorithm does not present many differences.

---

> ### Author Rebuttal · Authors · 2025-07-29
>
> ## Main answer
>
> We thank the reviewer snkJ for their thoughtful and constructive feedback, and for recognizing the theoretical depth of our contribution (Theorems 3.1, 3.2, 3.5), the novelty of the gradient flow sampling extension, and the strong empirical results on DNA sequence generation.
>
> We appreciate the reviewer’s concerns regarding missing baselines, computational costs, and the perception of limited innovation beyond adapting known RL techniques. Below, we address each point in detail and explain the design choices that guided our evaluation and algorithmic development.
>
> > The baselines included in this work remain narrow. Critical missing comparisons include inference-time-only approaches like discrete SMC methods or newer masked diffusion fine-tuning techniques.
>
> We thank the reviewer for this valuable suggestion. For the DNA task, we compared SEPO against a wide range of methods, including guidance-based techniques (including a discrete diffusion CG method), direct reward optimization, and policy gradient approaches.
>
> We acknowledge that recent masked diffusion finetuning methods, such as *d1* [1], are promising. However, they rely on **task-specific masking strategies and conditional sampling**, which do not extend naturally to fully unconditional reward-driven finetuning, which is central to our DNA setting.
>
> In our case, finetuning begins from a fully noisy input and proceeds through denoising without any conditioning on prefix sequences or external context. In contrast, *d1*-style methods assume a structure of the form $\frac{q_t^\theta(x \mid q)}{q_t^{\theta'}(x \mid q)}$, where $q$ is a "question" sequence sampled from a dataset. This implicitly enforces a conditional generation structure that is incompatible with our formulation. As a result, such baselines could not be included in this specific task setup.
>
> We will clarify this distinction in the revised version.
>
> Concerning discrete SMC, to the best of our knowledge, the most relevant work is [2], which appeared on arxiv after the NeurIPS submission deadline.
>
> > For language modeling, standard RLHF baselines, such as DPO or PPO-tuned transformers, are almost a must-include for a fairer evaluation.
>
> As for language modeling, we agree that including PPO-tuned transformers or DPO in the comparison would enrich the evaluation. However, these methods are typically applied to autoregressive models, and adapting them to the score-based diffusion setting is non-trivial. At the time of submission, these baselines had not yet been adapted to the score-based diffusion setting [3,4], and our focus for that experiment was primarily on demonstrating the versatility of SEPO.
>
> In fact, the language modeling experiment was intended as a **toy illustration** to show that SEPO is not limited to unconditional generation (as in DNA), but can also be applied to **standard conditional generation tasks**. The goal was to validate the general applicability of our method, rather than to benchmark against heavily optimized RLHF baselines designed for autoregressive models. We will make this intention clearer in the revised version.
>
> > SEPO’s inner sampling and score estimation steps (especially SNIS) introduce notable computational overhead; relevant discussions are not included.
>
> We agree with the reviewer that SNIS introduces additional computational overhead. As also noted by Reviewer 3dgX, we will include a dedicated discussion in the revised version detailing the wall-clock time, trade-offs involved, including compute scaling with sample count $M$, block size $n$, and the impact on training efficiency.
>
> It is true that SNIS involves additional computation due to the need to compute transition probabilities. Specifically, for each of the $N$ initial samples $(x_1, \ldots, x_N)$, we perform a batched forward pass through the score network to evaluate the transition probabilities required for SNIS. This leads to a moderate increase in compute.
>
> To quantify this, we now include wall-clock training time comparisons between SEPO and DRAKES in the following table (included in the revised version). These measurements were taken using the same hardware (a single GPU NVIDIA GeForce RTX 3090 with 24GB VRAM) and batch sizes to ensure fair comparison.
>
> | Method                | KL Trunc. Step | # Epochs | Batch Size/GRPO batch size | Runtime (hh:mm) | Time/Epoch (hh:mm:ss) | Final Pred-Activity |
> |------------------------|-------------|-----------|-------------|---------|-------------|----------------|
> | SEPO                   | 10          | 14       | 8 / 8       | 00:19   | 00:01:21    | 7.55         |
> | SEPO (gradient flow)   | 10          | 8       | 8 / 8       | 00:24   | 00:02:59    | 7.64         |
> | SEPO (no SNIS)         | 10          | 199       | 8 / 8        | 00:10   | 00:00:03    | 4.66          |
> | Drakes (no KL)         | -           | 127       | 8 / -       | 00:26   | 00:00:12    | 6.41          |
> | Drakes (with KL)       | 50          | 66       | 8 / -      | 00:31   | 00:00:28    | 5.70         |
>
> Even with this limited compute, it was possible to finetune models in a reasonable time.
>
> > Technically there is a concern with the technical hardness of this work. Compared to existing works, the major work is done by replacing continuous diffusion models with discrete diffusion models and the optimization algorithm does not present many differences.
>
> We thank the reviewer for raising this point and welcome the opportunity to clarify the technical contributions of our work. While SEPO draws conceptual inspiration from classical policy optimization methods (e.g., TRPO/PPO), **it is not a direct adaptation**. Instead, it introduces a new policy gradient framework tailored to discrete score-based diffusion, which presents unique algorithmic and theoretical challenges.
>
> In particular:
> - The core gradient estimator (Theorem 3.1) is made scalable using **self-normalized importance sampling (SNIS)**, which is crucial in the discrete setting. Unlike in continuous diffusion, where policy gradients can often be estimated via a single Monte Carlo sample, the discrete score-based gradient involves estimating itnractable marginals, making naive estimation infeasible. SNIS allows us to approximate this expectation in a principled way while maintaining asymptotically unbiasedness and variance control.
> - The method introduces a novel **score-based objective**, allowing REINFORCE-style learning under not only masked diffusion.
> - We also provide a **gradient flow sampling alternative**, which is technically non-trivial to define and implement in discrete spaces, and improves sample quality without relying on surrogate relaxations.
> - Finally, we prove a **convergence guarantee** (Theorem 3.5), which formalizes the stability of SEPO.
>
> We will better emphasize these contributions in the revised version, both in the introduction and technical sections.
>
>
> ## Answer to questions
>
> > Why are standard RLHF baselines such as PPO, DPO, or TRPO not included in the language modeling experiments, even though SEPO is proposed as a general-purpose policy gradient method?
>
> We answered this question above.
>
> > How critical is the gradient flow sampling mechanism to SEPO’s performance? Would simpler sampling strategies yield comparable results?
>
> Thank you for this question. The gradient flow sampling variant improves sample quality and stability (as seen in the DNA task), but it is **not** essential to the core algorithm. SEPO without this variant still outperforms baselines (see “SEPO” in Table 2). Gradient flow offers an orthogonal enhancement to SEPO that can be enabled depending on task constraints, at a higher complexity cost.
>
> We will clarify this point and report additional numbers in Appendix D.
>
> > Can the authors explain what the major difficulty of this algorithm (theoretically and experimentally) is beyond adapting existing policy optimization algorithms to a discrete-time formulation>
>
> We appreciate this important question. The main challenges of SEPO, that makes it more than a discrete-space adaptation to existing formulations, lie in:
>
> - **Gradient estimation**: Discrete diffusion models require estimating gradients under intractable conditional marginals (that are trivially adressed via vanilla Monte-Carlo estimations in the continuous case), which SNIS resolves under theoretical guarantees.
> - **Circumventing likelihood estimation** [1,5] by relying directly on the score network to develop a robust policy optimization method.
> - **General applicability**: Unconditional and conditional generation enabled, making SEPO a good tool if we just have access to a reward function and the score network. This is not necessarily the case with other methods such as [1,5].
>
> We will better highlight these technical challenges and the associated contributions in the revised version.
>
> [1] d1: Scaling Reasoning in Diffusion Large Language Models via Reinforcement Learning, Zhao et al. (2025)
>
> [2] Test-Time Alignment of Discrete Diffusion Models with Sequential Monte Carlo, Pani et al. (2025)
>
> [3] Discrete Diffusion Trajectory Alignment via Stepwise Decomposition, Han et al (2025)
>
> [4] Preference-based alignment of discrete diffusion models, Umberto et al. (2025)
>
> [5] wd1: Weighted Policy Optimization for Reasoning in Diffusion Language Models, Tang et al. (2025)

---

> > ### Comment · Reviewer_snkJ · 2025-08-05
> >
> > I thank the authors for the additional clarifications. After reading the response, most of my concerns have been addressed.However I am not fully convinced that "While SEPO draws conceptual inspiration from classical policy optimization methods (e.g., TRPO/PPO), it is not a direct adaptation. Instead, it introduces a new policy gradient framework tailored to discrete score-based diffusion, which presents unique algorithmic and theoretical challenges." To me, such transfer is straightforward, and the observation from this is not so unique.In a word, I will keep my current rating, leaning towards acceptance.

---

> > > ### Author Response · Authors · 2025-08-06
> > >
> > > We are very pleased that our clarifications have addressed the reviewer’s remaining concerns and that they appreciate the strengths of our work. We are especially grateful for the reviewer’s recognition of the paper’s contributions and for their willingness to lean toward acceptance. We thank the reviewer once again for their thoughtful and constructive review.

---

### Official Review · Reviewer_3dgX · 2025-07-03

**Clarity:** 4
**Significance:** 3
**Originality:** 3
**Rating:** 5
**Confidence:** 2

**Summary:**

The core algorithmic contribution is score entropy policy optimization (SEPO), which adapts policy gradient methods, commonly used in continuous RLHF, to discrete diffusion models. This addresses a significant challenge in fine-tuning these models, especially with non-differentiable reward functions, which are often encountered in practical applications like RLHF. SEPO provides an explicit characterization of policy gradient algorithms within the concrete score framework, supporting both conditional and unconditional generation. A technical innovation is the use of self-normalized importance sampling (SNIS) to estimate otherwise intractable marginal distributions required for gradient computation, which is claimed to offer consistent, asymptotically unbiased, and lower-variance estimates. The algorithm also incorporates concepts from trust region policy optimization (TRPO) to ensure stability and uses a clipping mechanism for policy updates. An alternative using gradient flow is also explored to improve sample quality.

**Questions:**

Theorem 3.1 estimates \(\nabla_\theta \ell_t(\theta)\) via SNIS, but how does variance scale with \(M\) (samples) and block size \(n\)? Fig 1 suggests instability for \(n \gg 10\)—was this tested?

Eq 4 includes KL regularization, yet experiments (Table 2) use \(\alpha=0.05\) without ablation. Why is KL critical for DNA but omitted in language tasks (Sec 4.1)?

Can the authors report GPU hours/memory for SEP0 vs. baselines (e.g., DRAKES) in Tables 1–2 to quantify efficiency claims?

**Ethical Concerns:**

["NO or VERY MINOR ethics concerns only"]

**Final Justification:**

I thank the authors for their detailed responses. I am convinced by their novelty arguments and their comments on scalability. The point concerning lack of baselines still stands as far as I am concerned. I have increased my scores for Clarity and Originality to reflect this.

**Limitations:**

Yes.

**Paper Formatting Concerns:**

No concerns.

**Quality:**

3

**Strengths And Weaknesses:**

Strengths:

1. Algorithmic innovation: SEP0 effectively bridges policy gradients with discrete diffusion, enabling non-differentiable reward optimization—a notable advance over Gumbel-Softmax alternatives.

2. Theoretical foundation: Convergence guarantees (Theorem 3.5) and gradient flow interpretation (Lemma 3.3) rigorously formalize the method.

3. Empirical rigor: Strong benchmarks in DNA/language tasks (Tables 1–2), with ablation confirming SNIS necessity (Table 8).

4. Novelty: SEP0 largely repurposes TRPO/PPO mechanics (Eq. 6–8), but in their rebuttal the authors convinced me that the unified framework adds sufficient innovation over prior RL-guided diffusion.  (Moved from Weakness to Strengths after rebuttal plus promised revisions.)

---

Weaknesses:

1. Incomplete baselines: Omits comparison to RLHF-tuned AR models (e.g., Llama 2) in language tasks. Does this undermine claims of diffusion superiority? Added after rebuttal: This is still a weak point of the paper.

---

I originally signalled scalability omissions as a weakness: No analysis of computational overhead from SNIS (Sec 3.1), especially for large \(n\) (e.g., \(m^n\) states); wall-clock times were also absent.  Added after rebuttal: In their rebuttal addressed this point in an acceptable manner.

---

> ### Author Rebuttal · Authors · 2025-07-29
>
> ## Main answer
>
> We thank Reviewer 3dgX for their detailed and thoughtful review, as well as for the positive assessment of our work. We are especially grateful for the recognition of the algorithmic contribution of SEPO, its theoretical guarantees, and its empirical performance on DNA and language modeling tasks.
>
> We appreciate the insightful questions and constructive suggestions, and we address each of these points below.
>
> > Novelty gap: SEP0 largely repurposes TRPO/PPO mechanics (Eq. 6–8); the "unified framework" adds limited innovation over prior RL-guided diffusion.
>
> We thank the reviewer for their comment. While SEPO is indeed inspired by the clipping mechanism of PPO/TRPO (which have proven to be robust and effective in a variety of reinforcement learning settings), we would like to emphasize that **our contribution is not a mere adaptation** of standard policy gradient methods to the diffusion setting.
>
> Instead, SEPO provides a principled and unified framework for **policy-based finetuning of discrete score-based diffusion models**, supporting both conditional and unconditional generation, and enabling optimization under non-differentiable rewards. To the best of our knowledge, **no prior work has formally derived policy gradients in this score-based discrete diffusion context**, nor has combined SNIS-corrected computation to reduce the variance with trust-region principles in this setting.
>
> We agree that the update rule (Eq. 6) bears resemblance to PPO, but we believe the theoretical formulation of the gradient (Theorem 3.1), its connection to concrete scores, and its integration with self-normalized importance sampling to manage intractable quantities (a bottleneck specific to the discrete case) constitutes a non-trivial extension. We will make these distinctions more explicit in the final version.
>
> > Scalability omissions: No analysis of computational overhead from SNIS (Sec 3.1), especially for large (n) (e.g., (m^n) states); wall-clock times absent.
>
> We thank the reviewer for this important remark. It is true that SNIS introduces some computational overhead. However, it is important to note that, in our setup, we never enumerate over the full $m^n$ state space. Thanks to the structure of the score network in discrete diffusion, we compute scores only over a small set of neighbors (i.e. states at Hamming distance 1, $\mathcal{O}(mn)$ states), not over the full support. This makes the gradient computation tractable even for moderate block sizes $n$.
>
> That said, SNIS still involves additional computation due to the need to compute transition probabilities. Specifically, for each of the $N$ initial samples $(x_1, \ldots, x_N)$, we perform a batched forward pass through the score network to evaluate the transition probabilities required for SNIS. This leads to a moderate increase in compute.
>
> To quantify this, we now include wall-clock training time comparisons between SEPO and DRAKES in the following table (included in the revised version). These measurements were taken using the same hardware (a single GPU NVIDIA GeForce RTX 3090 with 24GB VRAM) and batch sizes to ensure fair comparison.
>
> | Method                | KL Trunc. Step | # Epochs | Batch Size/GRPO batch size | Runtime (hh:mm) | Time/Epoch (hh:mm:ss) | Final Pred-Activity |
> |------------------------|-------------|-----------|-------------|---------|-------------|----------------|
> | SEPO                   | 10          | 14       | 8 / 8       | 00:19   | 00:01:21    | 7.55         |
> | SEPO (gradient flow)   | 10          | 8       | 8 / 8       | 00:24   | 00:02:59    | 7.64         |
> | SEPO (no SNIS)         | 10          | 199       | 8 / 8        | 00:10   | 00:00:03    | 4.66          |
> | Drakes (no KL)         | -           | 127       | 8 / -       | 00:26   | 00:00:12    | 6.41          |
> | Drakes (with KL)       | 50          | 66       | 8 / -      | 00:31   | 00:00:28    | 5.70         |
>
> Even with this limited compute, it was possible to finetune models in a reasonable time.
>
> > Incomplete baselines: Omits comparison to RLHF-tuned AR models (e.g., Llama 2) in language tasks. Does this undermine claims of diffusion superiority?
>
> The language modeling experiment was intended as a **toy illustration**, to demonstrate that SEPO can be applied beyond unconditional settings like DNA generation and is compatible with **standard conditional generation tasks**.  We agree that including comparisons with RLHF-tuned autoregressive models (e.g., PPO-tuned LLaMA-2, DPO) would strengthen the evaluation.
>
> However, our focus in this section was on showcasing the versatility of the SEPO framework, not making a claim about diffusion models' superiority in standard language modeling benchmarks. We will make this intention clearer in the revised version.
>
> ## Answer to questions
>
> > Theorem 3.1 estimates (\nabla_\theta \ell_t(\theta)) via SNIS, but how does variance scale with (M) (samples) and block size (n)? Fig 1 suggests instability for (n \gg 10)—was this tested?
>
> Fore the same reasons as above, we deal with $\mathcal{O}(mn)$ states in practice, instead of $\mathcal{O}(m^n)$.
>
> It can be shown (details will be added to the paper) that the variance of our SNIS estimation is $\text{Var}(\hat{q}^\theta_t(y)) = \mathcal{O}(\frac{1}{M}\text{Var}_{z\sim q^\theta_t(\cdot\mid y)}[\frac{1}{q^\theta_t(y\mid z)}])$. In the _worst case_ of a fully spread distribution $q^\theta_t(\cdot\mid y)$ over the $\mathcal{O}(mn)$ neighbours, the variance scale as $\text{Var}(\hat{q}^\theta_t(y)) = \mathcal{O}(\frac{n}{M})$. This does not grow exponentially with $n$, as potentially expected by the reviewer. Empirically, as expected, large values of $M$ tend to reduce the variance of the gradient estimation.
>
> > Eq 4 includes KL regularization, yet experiments (Table 2) use (\alpha=0.05) without ablation. Why is KL critical for DNA but omitted in language tasks (Sec 4.1)?
>
> KL regularization is useful for avoiding over-optimization. We used it in both the DNA and language experiments. We will clarify this for the language experiment in Appendix E.1 at the camera ready version.
>
> > Can the authors report GPU hours/memory for SEP0 vs. baselines (e.g., DRAKES) in Tables 1–2 to quantify efficiency claims?
>
> This question was adressed above.

---

### Official Review · Reviewer_WxVH · 2025-07-04

**Clarity:** 3
**Significance:** 4
**Originality:** 4
**Rating:** 5
**Confidence:** 4

**Summary:**

The authors propose the need for policy gradient methods for finetuning/training Discrete Diffusion moedels for Language modeling. In this paper the authors make the following contributions: (1) provide an explicit characterization of policy gradient algorithms for discrete diffusion models which allows for the use fo non-differentiable reqard function as well as conditional and unconditional generation, (2) propose a policy gradient algorithm called Score Entropy Policy Optimization (SEPO) with a gradient flow alternative and (3) validate the effectiveness of the SEPO family of algorithms with experiments on DNA finetuning and Language modeling

Main Method: the authors first describe the policy gradients for score entropy where they use the discrete REINFORCE trick to the score entropy changes and compute the loss function. To address teh high variance in REINFORCE, the authors build on using TRPO fundamentals to use an importance sampling term. This is then clipped in the interval $[1 - \epsilon, 1+\epsilon]$ to build the SEPO algorithm also described in Algorithm 1

**Questions:**

1. Maybe out of the scope of this work, but is there a workaround to allow for edits to the input without having to rerun the complete inference step?
2. For unconditional sampling, some examples of unconditional text generation or DNA sequence generation would add value to the paper

**Ethical Concerns:**

["NO or VERY MINOR ethics concerns only"]

**Final Justification:**

The authors have satisfactorily answered my clarification questions, and as I did not have any major concerns to begin with, I keep the same score.

**Limitations:**

yes

**Quality:**

4

**Strengths And Weaknesses:**

### Strengths
- The formulation works on the gaps of prior approaches such as DRAKES, GLIDE and d1 to handle unconditional generation, as well as non-diffferentiable rewards
- Rewards can be provided at any time step (l110) which allows for more flexible reward schemes and also validated in Table 1
- The paper introduces policy gradients in a principled manner while ensuring that the work is mostly self-contained with relevant works discussed in the main paper and the appendix
- Gradient Flow interpretation using $KL(p||\pi_\theta)$ demonstrates strong metrics on in the DNA sequence modeling task especially on Pred-Activity and ATAC-Acc. Furthermore, these scores have significantly lower variance demonstrating the stability of the training algorithm (Fig 2)
- The convergence bounds estimate of $\mathcal{O}(\frac{1}{\sqrt{S}})$ in Theorem 3.5 gives useful gurantees to the algorithm
- The authors validate the importance of

### Weaknesses
- In theory, unconditional sampling should be possible, but it is not empirically discussed.
- Specifically for language models, a window of tokens is typically predicted in Discrete Diffusion models. This makes it quite expensive to update the model outputs with small edits to the inputs or the prediction, hence reducing its scalability.

---

> ### Author Rebuttal · Authors · 2025-07-29
>
> ## Main answer
>
> We thank the reviewer WxVH for their detailed feedback, positive appreciation of our work and insightful evaluation. We are particularly grateful for the recognition of the theoretical contributions, the strong and stable performance of SEPO on experimental tasks, and the strong performance of the gradient flow formulation. We address their comments point by point below.
>
> > In theory, unconditional sampling should be possible, but it is not empirically discussed.
>
> Thank you for this remark. We would like to clarify that **unconditional generation is indeed implemented and evaluated** in our DNA finetuning experiments. In particular, our finetuning procedure for this DNA task starts from a fully noisy state and performs denoising via the learned policy, without conditioning on any prefix or reference sequence. The output is then evaluated through a reward function. This setup makes **SEPO truly applicable in an unconditional regime**, both in theory and in practice.
>
> This ability stems from a key design choice of SEPO: the policy gradient is computed solely based on the model’s outputs over arbitrary inputs, including fully noised ones, without requiring access to reference “questions” or partial conditioning (namely, we clip probability ratios of the form $\frac{q_t^{\theta}(x)}{q_t^{\theta'}(x)}$). This is not the case in other approaches such as d1 [1], where the policy gradient depends on probability ratios of the form $\frac{q_t^{\theta}(x\mid q)}{q_t^{\theta'}(x\mid q)}$ (with "questions" $q$ sampled from a dataset $\mathcal{D}$). These setups implicitly assume a conditional structure.
>
> While conditioning can be powerful when the goal is to precisely control outputs or target specific edits (e.g., finetuning within narrow constraints), unconditional finetuning offers a complementary strength: it enables the model to learn global trends or broad improvements over the generative process, such as enhancing biological activity in DNA sequences without anchoring to any predefined sequence. **This makes SEPO particularly appealing for general-purpose reward-driven adaptation**, and we believe this is one of its most promising capabilities.
>
> > Specifically for language models, a window of tokens is typically predicted in Discrete Diffusion models. This makes it quite expensive to update the model outputs with small edits to the inputs or the prediction, hence reducing its scalability.
>
> We thank the reviewer for this insightful comment, which we address below in question 1.
>
> ## Answer to questions
>
> > Maybe out of the scope of this work, but is there a workaround to allow for edits to the input without having to rerun the complete inference step?
>
> This is a very relevant and insightful question. While discrete diffusion models typically involve a full denoising process from noise to signal, the flexibility of our policy-based formulation makes it entirely possible to explore localized or partial denoising strategies. For instance, one could choose to modify only a subset of positions in the input sequence, and running just the end of the inference. These kinds of strategies fall within the scope of our method's **conditional generation setting** (l169-173), and can be implemented without altering the core algorithm.
>
> Moreover, our framework also supports the assignment of rewards at intermediate denoising steps. As noted in lines 111–114, *"one can run fewer denoising steps and stop the process at some intermediate time $T_0 < T$, assigning the reward $R_{T_0} = R$ based on the partially denoised sample."* This provides additional flexibility in practice, enabling users to trade off inference cost and finetuning granularity as needed.
>
> We will make theses capabilities more explicit in the revised version by adding a dedicated discussion paragraph in the paper.
>
> > For unconditional sampling, some examples of unconditional text generation or DNA sequence generation would add value to the paper
>
> Thank you for this suggestion. In fact, our DNA finetuning experiments are conducted in a fully unconditional generation setting. The model starts from a fully noisy sequence and denoises it step-by-step, without any prefix or conditioning. We will clarify this point more explicitly in the revised version and include sample sequences from the model in the appendix to illustrate unconditional generation. For text, our focus was on conditional generation following standard practice, but we agree that including qualitative examples of unconditional samples would add value and will incorporate these as well. We will add some unconditional samples of generated text in Appendix E.1.
>
> [1] d1: Scaling Reasoning in Diffusion Large Language Models via Reinforcement Learning, Zhao et al. (2025)

---

> > ### Comment · Reviewer_WxVH · 2025-08-07
> >
> > Thank you to the authors for the detailed response.
> >
> > > We would like to clarify that unconditional generation is indeed implemented and evaluated in our DNA finetuning experiments.
> >
> > Thank you for the clarification. I had misunderstood the DNA sequence modeling experiments as conditional models, while they were actually finetuned using enhancer activity score in the HepG2 cell line.
> >
> > > one could choose to modify only a subset of positions in the input sequence, and running just the end of the inference.
> >
> > L169-173 mostly describe the conditional form of the Importance sampling of the outer sum (i.e. eq 6).  It is not immediately clear how that would help with "running just the end of the inference". While the reward can be set up for any/all number of intermediate steps of the diffusion process, how would one go about learning the reward functions without any supervision available for these intermediate steps. Could the authors expand a bit more in this direction?

---

> > > ### Author Response · Authors · 2025-08-08
> > >
> > > We thank the reviewer for the follow-up and for the opportunity to clarify this point.
> > >
> > > By ``running just the end of the inference'' here, we mean fixing certain positions in the sequence (e.g., some words) and denoising only the unfixed ones. This allows starting the process from this partially denoised sequence and running $T$ denoising steps. At any step $t \in [0, \ldots, T]$, a partially denoised sequence $x = x_t$ can still be evaluated by a _fixed_ reward function $R_t$, giving the $R_t(x)$ that appears in the gradient formulas. These reward functions are not assumed to be learnable: in our setup, they are more generally task-specific scoring functions (e.g., activity predictors for DNA) that assign a score $R_t(x)$ to $x=x_t$ regardless of whether it is fully denoised or partially noisy.
> > >
> > > We hope this clarification addresses the reviewer’s concern, and we thank them again for raising this point.

---

> > > > ### Comment · Reviewer_WxVH · 2025-08-08
> > > >
> > > > > By ``running just the end of the inference'' here, we mean fixing certain positions in the sequence (e.g., some words) and denoising only the unfixed ones.
> > > >
> > > > Makes sense, this appears to be somewhat in the direction of Masked Diffusion models with theory on how to incorporate step-level rewards in the reverse/forward diffusion.

---

### Author Response · Authors · 2025-08-06

As the rebuttal period draws to a close, we would like to thank all reviewers for their constructive feedback. We’re especially pleased that two reviewers (snkJ and sGKa) engaged in detailed discussion with us, and that one reviewer (sGKa) chose to raise their score.

With that in mind, we highlight the principal strengths of our work:

- We derive general discrete diffusion finetuning objective for any reward sequence $\{R_t\}$ (Theorems 3.1–3.2) and prove **convergence bounds** (Theorem 3.5).

- State-of-the-art empirical performance : Across DNA tasks, SEPO **outperforms strong and various baselines**, demonstrating the practical power of concrete score optimization for RL-based fientuning.

- Despite slightly higher per-epoch cost, SEPO requires an order of magnitude fewer epochs to reach peak performance, yielding a **lower overall wall-clock time** than competing methods.

- Our framework naturally supports conditional and unconditional generation and can readily incorporate meaningful intermediate rewards or bias-control techniques in future works, as discussed with reviewer sGKa.

We thank the reviewers once again for their time and thoughtful comments, and we hope these highlights will earn a strong endorsement of our manuscript.

---

### Decision · Program_Chairs · 2025-09-17

**Decision:**

Accept (poster)

**Comment:**

The paper proposed a policy gradient method tailored for discrete diffusion models. The proposed method extends the TRPO/PPO style policy gradient to discrete diffusion modeling and can handle non-differentiable rewards as well as rewards given in the intermediate steps. The experiments are conducted in two different domains, language and molecules, confirming the effectiveness of the proposed approach. During the discussion, the concerns regarding the computational overhead, the limited set of baselines, the technical novelty, and a technical flaw in the theoretical claim are raised and actively discussed. The concerns are mostly well addressed in the rebuttal. The authors are strongly encouraged to incorporate the comment regarding Theorems 3.1 and 3.2 from Reviewer sGKa in the updated manuscript. Despite that, the algorithmic and empirical contributions from this paper remain solid, and thus I recommend acceptance.